# Axis-level Reflectional Symmetry Detection with Group-Equivariant Representation

## Abstract

Reflectional symmetry detection remains a challenging task in machine perception, particularly in complex real-world scenarios involving noise, occlusions, and distortions. We introduce a novel equivariant approach to axis-level reflectional symmetry detection that effectively leverages dihedral group-equivariant representation to detect symmetry axes as line segments. We propose orientational anchor expansion for fine-grained rotation-equivariant analysis of diverse symmetry patterns across multiple orientations. Additionally, we develop reflectional matching with multi-scale kernels to extract effective cues of reflectional correlations, allowing for robust symmetry detection across different receptive fields. Our approach unifies axis-level detection with reflectional matching while preserving dihedral group equivariance throughout the process. Extensive experiments demonstrate the efficacy of our method while providing more accurate axis-level predictions than existing pixel-level methods in challenging scenarios.

## 1 Introduction

Symmetry is a fundamental concept prevalent in both natural and artificial environments (Wertheimer, 1938; Tyler, 1995), appearing at various scales and across different domains (Møller & Thornhill, 1998; Giurfa et al., 1996). Although human vision naturally recognizes symmetry(Wagemans, 1995), the task of detecting symmetry remains challenging for machine perception in real-world scenarios (Liu et al., 2010). This work focuses on reflectional symmetry, which is the most basic type of symmetry, also called mirror, bilateral, or line symmetry. Reflectional symmetry detection aims to identify and localize axes of symmetry from an image along which visual elements on one side are mirrored onto the other. The primary challenges of the task involve accurately identifying individual symmetry axes within complex, cluttered images where visual patterns often include noise, occlusions, and distortions, as well as precisely estimating the orientation and length of these axes.

Early approaches to symmetry detection relied on pairwise feature matching, where symmetry was identified by matching features reflected across the axis of symmetry (Loy & Eklundh, 2006; Cho & Lee, 2009; Cornelius et al., 2007; Sun & Si, 1999; Kiryati & Gofman, 1998). With the advent of deep learning and the availability of real-world datasets, convolutional neural network (CNN)-based models significantly advanced symmetry detection. Early work by Gens & Domingos (2014) proposed one of the first equivariant architectures for handling symmetries in CNNs, paving the way for more sophisticated approaches. Subsequently, Funk & Liu (2017) exploited deep appearance features to predict pixel-level reflectional symmetry axis. PMCNet (Seo et al., 2021) matched pixels across potential axes in the spatial dimension, incorporating self-similarity measures for symmetry detection. Meanwhile, EquiSym (Seo et al., 2022) introduced an equivariant network capable of producing reflection and rotation-equivariant symmetry detection predictions. Recent work has further expanded the scope of symmetry detection. Recent works have expanded symmetry detection across diverse fields: (Zhang et al., 2023) proposed 3D reflection symmetry prediction from depth images, (Podgorelec et al., 2023) applied it to satellite imagery, and (Hojny, 2024) developed techniques for handling symmetries in optimization problems.

Despite these advancements, existing neural approaches primarily frame 2D reflectional symmetry detection as a pixel-level heat-map prediction problem (Funk & Liu, 2017; Seo et al., 2021; 2022). Given that an axis of reflectional symmetry is a line segment with a specific orientation and length,

these methods fall short in delivering precise axis-level symmetry information. Furthermore, while the detection of symmetry axes requires consistent recognition of patterns in arbitrary rotation and reflection, existing neural approaches do not fully consider the structure of symmetry in both feature representation and output prediction, which limits the performance in accuracy and consistency.

To address the aforementioned limitations, we introduce a novel equivariant approach for 2D axis-level reflectional symmetry detection that extensively leverages dihedral group-equivariant representation. We represent an axis of reflectional symmetry as a line segment with its midpoint, length, and orientation and perform axis-level prediction from dense feature map equivariant to reflection, rotation, and translation. To effectively leverage the group-equivariant features into axis detection, we propose to create orientational anchors along the dimension of rotation and incorporate reflectional matching along each orientation that calculates correlations across potential symmetry axes. The reflectional matching module is further enhanced with multi-scale kernels to capture symmetries across various receptive fields. The resulting method unifies orientational anchor-based axis-level detection with reflectional matching and enables more robust and comprehensive symmetry detection while preserving group equivariance throughout the process.

Through extensive experiments, we demonstrate that our axis-level detection approach, combined with the orientational anchor and the reflectional matching module based on multi-scale kernel, leads to significant performance gains across various challenging scenarios, consistently outperforming existing pixel-level benchmarks. The key contributions of this paper include:

- We introduce a novel axis-level reflectional symmetry detection network that extensively leverages dihedral group-equivariant representation.
- We propose orientational anchor expansion that enables fine-grained rotation-equivariant analysis of diverse symmetry patterns across multiple orientations.
- We develop reflectional matching that extracts effective cues of reflectional correlations with multi-scale kernels for robust symmetry detection across different receptive fields.
- We validate the effectiveness of our proposed components, demonstrating superior performance over existing methods in real-world scenarios.

## 2 RELATED WORK

### 2.1 REFLECTIONAL SYMMETRY DETECTION

Reflectional symmetry detection has evolved from using hand-crafted features like contours, edges and histogram (Shen et al., 2001; Atadjanov & Lee, 2016; Prasad & Davis, 2005; Wang et al., 2014; 2015; Sun & Si, 1999; Cornelius et al., 2007), and image gradients (Gnutti et al., 2021; Sun & Si, 1999; Kiryati & Gofman, 1998) to leveraging deep learning techniques (Fukushima & Kikuchi, 2006; Funk & Liu, 2017; Seo et al., 2021; 2022). Early approaches used SIFT descriptors Lowe (2004) for orientation determination (Loy & Eklundh, 2006) and feature matching (Cho & Lee, 2009). Recent advancements include CNN-based methods (Funk & Liu, 2017), polar self-similarity descriptors (Seo et al., 2021), and group-equivariant neural networks (Seo et al., 2022). However, these methods treat symmetry detection as a point-wise detection($i.e.$, heatmap-prediction) task, limiting their ability to accurately localize individual symmetry axes. Our work addresses this limitation by reframing symmetry detection as a axis-wise detection problem, enabling detailed analysis of symmetry axes.

### 2.2 EQUIVARIANT NEURAL NETWORKS

Convolutional neural network has been a breakthrough in deep learning with its inductive bias and parameter sharing via translation equivariance. Afterward, there has been decades of studies to extend the equivariance of the CNNs to the other symmetry groups such as rotation and reflection for robust image recognition.

Group equivariant CNNs Cohen & Welling (2016a;b) pioneered this field, building on Gens & Domingos (2014)'s early work on symmetry-aware architectures. Subsequent research explored more specific types of equivariance, with Dieleman et al. (2015) demonstrating the effectiveness of rotational invariance through galaxy morphology prediction. These works laid the foundation for

further advancements including circular harmonics Worrall et al. (2017), vector fields Marcos et al. (2017), and hexagonal lattices Hoogeboom et al. (2018). Recent studies have comprehensively explored equivariant CNNs in terms of homogeneous spaces and groups Weiler et al. (2018); Cohen et al. (2018b;a; 2019); Weiler & Cesa (2019). Redet Han et al. (2021) utilized cyclic-group CNNs for aerial object detection but encountered equivariance breaks due to stride-2 group equivariant convolutions. Similarly, EquiSym Seo et al. (2022) applied $D_8$-equivariant networks for symmetry detection but also faced equivariance breaks due to stride-2 group convolutions and lacked explicit feature matching across symmetry axes. In contrast, our approach maintains full equivariance for $C_N$ group and incorporates feature matching to detect reflectional symmetry more accurately.

## 2.3 Line segment detection

Axes are usually represented with line segments. Accurate detection of line segments is directly linked to axes detection via their orientation, length and position. Early line detection techniques evolved from edge detection methods, leveraging gradients to identify linear features (Canny, 1986; Deriche, 1987; Lu et al., 2015). Subsequent approaches focused on post-processing gradient-derived features, including connecting edges (Akinlar & Topal, 2011), concatenating edge fragments (Cho et al., 2018), region-growing Grompone von Gioi et al. (2010), and Hough transformation Xu et al. (2015). Recent deep learning-based methods have adapted object detection principles to line segment detection. Notable contributions include an indoor dataset Huang et al. (2018), the Structural Average Precision metric Zhou et al. (2019), 4D attraction fields Xue et al. (2023), and transformer-based approaches Xu et al. (2021). Alternative representations such as tri-point Huang et al. (2020) and Center-Angle-Length (CAL)Zhang et al. (2021) have also been proposed. However, these methods use a global anchor approach, predicting orientation offsets from a single anchor line. This approach struggles with overlapping lines of different orientations and has a large orientation prediction range. We address these limitations by introducing a Center-Angle-Length (CAL)-based orientational anchor using multiple anchor lines, improving detection of overlapping lines and reducing the orientation prediction range.

## 3 Background

### 3.1 Groups, Symmetry, and Equivariance

A group is a mathematical structure consisting of a set and an operation that satisfies four fundamental properties: closure, associativity, identity, and invertibility (Rotman, 2012). Groups are particularly useful for describing symmetries—transformations such as rotations or reflections that leave an object invariant. For example, the cyclic group $C_N$ represents the set of discrete rotations $\{r^0, \ldots, r^{N-1}\}$, where $r$ is the generator for rotation. The group law for cyclic rotations is given by $r^i r^j = r^{(i+j) \bmod N}$. The dihedral group $D_N$, which is especially relevant for our purposes, incorporates both rotational and reflectional symmetries. It can be expressed as $D_N = \{r^0, r^1, \ldots r^{N-1}, b, br^1, \ldots br^{N-1}\}$, where $b$ and $r$ are generators for reflection and rotation, respectively. The group laws for $D_N$ are $b^2 = e$ and $r^n b = br^{-n}$, where $e$ is the identity element.

Equivariance refers to the property of a function $f : \mathcal{X} \to \mathcal{Y}$ that commutes with the action of a group $G$. Formally, for linear group representations which map groups to linear spaces, $\sigma_1 : G \to \mathrm{GL}(\mathcal{X})$ and $\sigma_2 : G \to \mathrm{GL}(\mathcal{Y})$, the function $f$ is equivariant if:

$$f(\sigma_1(g) \cdot x) = \sigma_2(g) \cdot f(x), \quad \forall g \in G, x \in \mathcal{X}. \tag{1}$$

In neural networks, equivariance ensures that input transformations lead to predictable and structured transformations in the output, preserving symmetries in data.

### 3.2 Group Representation

A group representation maps each element of a group $G$ to a linear transformation of a vector space Cohen & Welling (2016b); Weiler et al. (2018). One important type of representation is the regular representation Cohen & Welling (2016a), which acts on the vector space $\mathbb{R}^{|G|}$, where $|G|$ is the order of the group.

For a finite group $G = \{g_1, \ldots, g_N\}$, the regular representation $\sigma_{\mathrm{reg}}^G(g)$ for any element $g \in G$ is defined as:

$$\sigma_{\mathrm{reg}}^G(g) = [\mathbf{e}_{gg_1}, \ldots, \mathbf{e}_{gg_N}], \tag{2}$$

where $\mathbf{e}_{g_i} \in \mathbb{R}^{|G|}$ is a standard basis vector corresponding to the group element $g_i \in G$. This representation permutes these basis vectors according to the group action. Notably, the regular representation of the identity element $e \in G$ is the identity matrix: $\sigma_{\mathrm{reg}}^G(e) = \mathrm{I}_{|G|}$. For the cyclic group $\mathrm{C}_N$, the regular representation of an element $r^n \in \mathrm{C}_N$ is a cyclic permutation matrix:

$$\sigma_{\mathrm{reg}}^{\mathrm{C}_N}(r^n) = \left[ \mathbf{e}_{r^n}, \mathbf{e}_{r^{(n+1) \bmod N}}, \ldots, \mathbf{e}_{r^{(n+N-1) \bmod N}} \right]. \tag{3}$$

For the dihedral group $\mathrm{D}_N$, which has $2N$ elements, the regular representation for a general element $r^n b$ permutes both rotational and reflectional symmetries of the group:

$$\sigma_{\mathrm{reg}}^{\mathrm{D}_N}(br^n) = \left[ \mathbf{e}_{br^n}, \mathbf{e}_{br^{(n+1) \bmod N}}, \ldots, \mathbf{e}_{br^{(n+N-1) \bmod N}}, \right.$$
$$\left. \mathbf{e}_{r^n}, \mathbf{e}_{r^{(n+1) \bmod N}}, \ldots, \mathbf{e}_{r^{(n+N-1) \bmod N}} \right]. \tag{4}$$

This representation captures the structure of $\mathrm{D}_N$ in a $2N \times 2N$ matrix.

### 3.3 GROUP CONVOLUTION

To generalize standard convolution to handle group symmetries, we use group convolutions Cohen & Welling (2016a). For a lifted feature map $f_G$, which associates each spatial position with a group element, group convolution is defined as:

$$(f_G *_G \psi)(g, \mathbf{x}) = \sum_{g' \in G} \sum_{\mathbf{y} \in \mathbb{Z}^2} f_G(g', \mathbf{y}) \psi(g^{-1}(\mathbf{x} - \mathbf{y}))(g'). \tag{5}$$

Here, $\psi$ is the group convolution filter, $f_G$ is the lifted feature map, and $g, g' \in G$ are group elements. The key property of group convolution is its equivariance, meaning that applying a group transformation to the input results in a corresponding transformation in the output:

$$[(g' \cdot f_G) *_G \psi](\mathbf{x}) = [g' \cdot (f_G *_G \psi)](\mathbf{x})$$
$$= \sigma_{\mathrm{reg}}^G(g') \cdot (f_G *_G \psi)(g'^{-1} \cdot \mathbf{x}). \tag{6}$$

This ensures that symmetries are preserved throughout the convolutional layers. Detailed explanations about the regular representation and group convolution are provided in the A.

## 4 PROPOSED METHOD

We introduce a group-equivariant neural network for axis-level reflectional symmetry detection that effectively learns to detect axes of reflectional symmetry patterns from an image. While previous approaches such as Dieleman et al. (2015) handle rotation invariance through input transformation and ensemble strategies, our method takes a more general and principled approach by using a dihedral group-equivariant neural network for 2D feature extraction Cohen & Welling (2016a), which mathematically guarantees equivariance to both rotations and reflections. Basically, the network is designed to classify the presence of a mid-point of a reflectional symmetry axis for each pixel position and also regress the angle and length of the axis (Sec. 4.1).

To effectively leverage the equivariant representation for reflectional symmetry detection, we create anchor lines over orientational dimension (Sec. 4.2), introduce equivariant reflectional matching that computes reflectional correlations across each anchor line (Sec. 4.3), and expand the equivariant reflectional matching to multi-scales (Sec. 4.4). Note that all these modules are designed to be dihedral group-equivariant so that the network provides consistent axis predictions over rotation and reflection. The overall architecture of our method is illustrated in Fig. 1.

### 4.1 AXIS-LEVEL REFLECTIONAL SYMMETRY DETECTION

While existing neural approaches to reflectional symmetry (Funk & Liu, 2017; Seo et al., 2021; 2022) all aim to predict a pixel-level heat-map for the presence of symmetry axes at each position, a more complete and direct approach is to detect reflectional symmetry axes as line segments that specify the angle, length, and position of each symmetry axis. Inspired by recent neural networks for line detection Zhang et al. (2021), we propose an axis-level reflectional symmetry detection network that predicts individual symmetry axes as line segments as follows.

We build a $\mathrm{D}_8$-equivariant residual network based on ResNet-34 (He et al., 2016). We address the issue of broken equivariance caused by stride-2 group convolution Cohen & Welling (2016a), as

Figure 1: **Overall pipeline of our proposed reflectional symmetry detection method.** The pipeline consists of: (a) $D_N$-equivariant backbone for feature extraction (top left), (b) Reflectional matching module for computing similarity scores across rotations and reflections (bottom, blue), (c) Multi-scale matching which employs multi-scale kernels (middle left, pink), (d) $C_N$-equivariant branch and (e) Orientational axis reconstruction (middle right, green). This architecture maintains equivariance throughout the entire process, enabling efficient and precise symmetry detection across various orientations.

used in existing networks Han et al. (2021) for downsampling. To deal with, we replace stride-2 convolutions with a stride-1 convolution followed by max pooling.

Given an input image, the base feature map $\mathbf{F}$ is extracted from our equivariant network backbone. This feature map is then passed through detection branch $\mathcal{B}$ to produce the output $\mathbf{Y}$:

$$\mathbf{Y} = \mathcal{B}(\mathbf{F}) \in \mathbb{R}^{H \times W \times 3}. \tag{7}$$

The three output channels correspond to the mid-point score $p$, line length $\rho$, and orientation $\theta$, respectively. At each position $(x, y)$, the output $\mathbf{O}_{(x,y)} = (p, \rho, \theta)$ is obtained. A line at $(x, y)$ is represented as $\mathbf{o} = (x, y, \rho, \theta)$. The start and end points are given by:

$$\begin{bmatrix} x^s \\ y^s \end{bmatrix} = \begin{bmatrix} x \\ y \end{bmatrix} + \frac{\rho}{2} \begin{bmatrix} \cos(\theta) \\ \sin(\theta) \end{bmatrix}, \tag{8}$$

$$\begin{bmatrix} x^e \\ y^e \end{bmatrix} = \begin{bmatrix} x \\ y \end{bmatrix} - \frac{\rho}{2} \begin{bmatrix} \cos(\theta) \\ \sin(\theta) \end{bmatrix}. \tag{9}$$

Since our training objective consists of loss terms for mid-point classification, length regression, and orientation regression. The mid-point classification loss uses weighted binary cross-entropy with weight $\gamma$:

$$\mathcal{L}_{\text{mid}} = \mathbb{E}_{(\alpha,x,y)} \left[ -\gamma p \log(\hat{p}) - (1-p) \log(1-\hat{p}) \right]. \tag{10}$$

The regression losses for length $\rho$ and orientation $\theta$ are applied only when the ground truth mid-point is valid ($p = 1$), as enforced by the indicator function $\mathbb{I}_{p=1}$:

$$\mathcal{L}_\rho = \mathbb{E}_{(\alpha,x,y)}[\mathbb{I}_{p=1} \cdot \text{SmoothL1}(\rho, \hat{\rho})], \tag{11}$$

$$\mathcal{L}_\theta = \mathbb{E}_{(\alpha,x,y)}[\mathbb{I}_{p=1} \cdot |\theta - \hat{\theta}|]. \tag{12}$$

The overall objective is the weighted sum of the individual terms:

$$\mathcal{L}_{\text{total}} = \mathcal{L}_{\text{mid}} + \lambda_\rho \mathcal{L}_\rho + \lambda_\theta \mathcal{L}_\theta. \tag{13}$$

While this base network is a lightweight adaptation of the existing line detection network Zhang et al. (2021) to symmetry-axis detection, the use of dihedral group-equivariant representation Cohen & Welling (2016a) improves it to produce more consistent prediction over rotation and/or flip transformation, which ensures robustness in dealing with arbitrary orientations of the symmetry axes.

## 4.2 Orientational Anchor Expansion

The axis detection network above can be seen as generating axes by placing an anchor line, which is analogous to an anchor box in object detection Ren (2015); Liu et al. (2016), for each position on the translational dimension of the feature map, *i.e.* $x$-$y$ position, and then assigning an angle displacement and length to the anchor lines. Now that our group-equivariant feature map has an additional group dimension for rotation, the set of anchor lines can naturally be expanded into the rotational dimension. This enables our model to better utilize equivariant representations, enhancing its ability to predict axis orientations. To this end, we expand the output $\mathbf{Y}$ for $|\mathrm{C}_N|$ orientational anchors as:

$$\mathbf{Y} = \mathcal{B}(\mathbf{Z}) \in \mathbb{R}^{|\mathrm{C}_N| \times H \times W \times 3}, \tag{14}$$

where the last dimension corresponds to the mid-point score $p$, line length $\rho$, and orientation $\theta$. Aggregation across cyclic group components is performed as:

$$\mathbf{O}_\alpha = \mathbf{Y}_\alpha + \mathbf{Y}_{\alpha + N/2}, \quad \alpha = 1, \dots, \tfrac{N}{2}, \tag{15}$$

where $\alpha$ is the channel index, and $\mathbf{O} \in \mathbb{R}^{|\mathrm{C}_N|/2 \times H \times W \times 3}$ remains $\mathrm{C}_N$-equivariant when repeated across the channel dimension. This aggregation accounts for the equivalence of orientations $\theta$ and $\theta + \pi$, ensuring symmetrical orientations are handled consistently. Each $\mathbf{O}_\alpha$ represents an anchor capturing orientation offsets within the range $[-\frac{\pi}{N}, \frac{\pi}{N})$ from its corresponding orientation, $\frac{2\pi\alpha}{N}$. Orientation offsets are predicted rather than absolute orientations to maintain rotational equivariance. While absolute orientations change with rotation, the length, orientation offset, and mid-point probability remain invariant. At each position $(\alpha, x, y)$, the output is $\mathbf{O}_{(\alpha, x, y)} = (p, \rho, \theta)$. A line at $(\alpha, x, y)$ is represented as $\mathbf{o} = (\alpha, x, y, \rho, \theta)$. The start and end points of the line are computed as:

$$\begin{bmatrix} x_\alpha^s \\ y_\alpha^s \end{bmatrix} = \begin{bmatrix} x_\alpha \\ y_\alpha \end{bmatrix} + \frac{\rho}{2} \begin{bmatrix} \cos(\theta_\alpha) \\ \sin(\theta_\alpha) \end{bmatrix}, \tag{16}$$

$$\begin{bmatrix} x_\alpha^e \\ y_\alpha^e \end{bmatrix} = \begin{bmatrix} x_\alpha \\ y_\alpha \end{bmatrix} - \frac{\rho}{2} \begin{bmatrix} \cos(\theta_\alpha) \\ \sin(\theta_\alpha) \end{bmatrix}, \tag{17}$$

where $\theta_\alpha = \frac{2\pi\alpha}{N} + \theta$. This approach greatly reduces missing detections from overlapping midpoints and restricts each anchor's search space to specific orientation offsets, enabling the model to efficiently detect and differentiate multiple symmetry axes even in complex scenes.

## 4.3 Reflectional matching

The most intuitive way of validating a reflectional symmetry pattern is to compare the pattern with its mirrored or reflectional counterpart along its axis of symmetry, *i.e.*, reflectional matching Loy & Eklundh (2006); Cho & Lee (2009). Unlike hand-crafted local descriptors such as SIFT (Lowe, 2004), conventional neural feature maps do not provide proper features for this purpose as they are not equivariant to rotation and reflection. By levaraging our dihedral group-equivariant features, we introduce a principled technique for reflectional matching, which provide a strong cue for the presence of reflectional symmetry patterns. Given a single fiber feature $\mathbf{f} \in \mathbb{R}^{C|\mathrm{D}_N|}$, where $\mathcal{C}$ is the number of channels and $|\mathrm{D}_N|$ represents the size of the dihedral group, the fiber is structured according to the dihedral group $\mathrm{D}_N$. This structure allows the fiber to undergo cyclic rotations and reflections through the group's representations. Specifically, the transformation of the fiber under $l$-reflections followed by $n$-rotations is expressed as:

$$\mathbf{f}^{(l,n)} = \bigoplus_{c=1}^{\mathcal{C}} \sigma_{\mathrm{reg}}^{\mathrm{D}_N}(b^l r^n) \mathbf{f}_c, \tag{18}$$

where $\mathbf{f}_c \in \mathbb{R}^{|\mathrm{D}_N|}$ represents the group-equivariant subset of the fiber, and $\mathbf{f} = [\mathbf{f}_1, \dots, \mathbf{f}_\mathcal{C}]^\top$. Here, $\sigma_{\mathrm{reg}}^{\mathrm{D}_N}(b^l r^n)$ represents the regular representation of $\mathrm{D}_N$ for $l$ reflections and $n$ rotations. The matrix pre-multiplying entire $\mathbf{f}$ is block-diagonal matrix in $\mathbb{R}^{\mathcal{C}|\mathrm{D}_N| \times \mathcal{C}|\mathrm{D}_N|}$, with each block $\mathbb{R}^{|\mathrm{D}_N| \times |\mathrm{D}_N|}$ is

a single-channel permutation, repeated $\mathcal{C}$ times. The reflectional similarity $h$ for two inputs $\mathbf{f}^1, \mathbf{f}^2 \in \mathbb{R}^{\mathcal{C}|\mathrm{D}_N|}$ is defined as:

$$h(\mathbf{f}^1, \mathbf{f}^2) = \bigoplus_{c=1}^{\mathcal{C}} \frac{\mathbf{f}_c^1 \cdot \mathbf{f}_c^2}{\|\mathbf{f}_c^1\|\|\mathbf{f}_c^2\|} \in \mathbb{R}^{\mathcal{C}}. \tag{19}$$

To capture reflectional symmetry across different orientations, similarity scores are computed for each rotation, both for the rotated and rotated-then-reflected fibers:

$$\mathbf{H_x} = \bigoplus_{n=0}^{|\mathrm{C}_N|-1} h(\mathbf{F}_\mathbf{x}^{(0,n)}, \mathbf{F}_\mathbf{x}^{(1,n)}) \in \mathbb{R}^{\mathcal{C}|\mathrm{C}_N|}, \tag{20}$$

where $\mathbf{F}_\mathbf{x}^{(0,n)}$ and $\mathbf{F}_\mathbf{x}^{(1,n)}$ represent the fiber at position $\mathbf{x}$ under the regular representation for $n$ rotations, with and without reflection, respectively. The resulting similarity score map $\mathbf{H} \in \mathbb{R}^{\mathcal{C}|\mathrm{C}_N|\times H \times W}$ is equivariant to the cyclic group $\mathrm{C}_N$. A detailed proof demonstrating the equivariance of reflectional matching is provided in Appendix B. Unlike Eq. (7), which relies solely on the base feature map $\mathbf{F}$, the reflectional matching feature $\mathbf{H}$ is concatenated with $\mathbf{F}$, enhancing the robustness of symmetry detection when passed to the detection branches.

### 4.4 MULTI-SCALE MATCHING

While single-fiber reflectional matching only matches at fixed $x$-$y$ coordinates, we expand our network to explore the neighborhood through spatial, rotational, and reflectional transformations to detect broader symmetries. This neighborhood is described by a set of 2D offset vectors $\mathcal{Q}_k$:

$$\mathcal{Q}_k = \left\{ (i,j) \,\Big|\, i,j \in \left\{ -\tfrac{(2k+1)-1}{2}, \ldots, \tfrac{(2k+1)-1}{2} \right\} \right\}, k \in \mathbb{N}. \tag{21}$$

To compute the similarity between the transformed feature maps, the similarities are summed across the neighborhood $\mathcal{Q}_k$ and over all group transformations:

$$\mathbf{H}_\mathbf{x}^{(k)} = \sum_{\mathbf{q} \in \mathcal{Q}_k} \bigoplus_{n=0}^{|\mathrm{C}_N|-1} h(\mathbf{F}_{\mathbf{x}+r^n(\mathbf{q})}^{(0,n)}, \mathbf{F}_{\mathbf{x}+br^n(\mathbf{q})}^{(1,n)}), \tag{22}$$

where $b^l r^n(\mathbf{q})$ denotes the spatially transformed offset due to $l$ reflections and $n$ rotations. When $k = 1$, the operation simplifies to reflectional matching along a single fiber. To detect symmetries at multiple scales, we employ different kernel sizes, allowing the model to capture patterns across various spatial extents. The final feature map, $\mathbf{Z}$, combines the base feature map $\mathbf{F}$ with the multi-scale reflectional similarity features $\mathbf{H}^{(k_1)}, \ldots, \mathbf{H}^{(k_M)}$. This combined feature map is then fed into the detection branch, ensuring a comprehensive and robust symmetry detection process that fully exploits the network's equivariant properties.

## 5 EXPERIMENTS

### 5.1 IMPLEMENTATION DETAILS

**Dataset.** Most existing reflectional symmetry detection datasets either lack diversity in reflection axes or are no longer available, limiting their usefulness for modern benchmarks (Cicconet et al., 2016; Liu et al., 2013; Funk & Liu, 2017; Seo et al., 2021). As a result, we use the DENDI dataset (Seo et al., 2022), which provides more diverse and comprehensive annotations, including multiple symmetry axes and continuous symmetry groups. For better generalization of the model, we use standard augmentations like flipping, rotation, and color jittering. Additionally, we extract 7k axis-annotated masks from the training set and paste them onto other images, avoiding overlap with existing annotations. By adding 1 to 6 objects per iteration and repeating the process three times, the dataset expands 18-fold to contain 30k training images.

**Evaluation metrics.** For axis-level reflectional symmetry evaluation, we use structural Average Precision (sAP) (Zhou et al., 2019) with an additional condition to accommodate non-line annotations in the DENDI dataset. In our evaluation, a predicted line $l_p$ is considered a

Table 1: Axis-level reflectional symmetry detection on the DENDI dataset.

| Method | sAP | | | sAP$_{\text{img}}$ | | |
|---|---|---|---|---|---|---|
| | @5 | @10 | @15 | @5 | @10 | @15 |
| Axis-level detection base network (Sec. 4.1) | 4.8 | 7.9 | 10.0 | 11.2 | 14.3 | 16.7 |
| + Orientational anchor expansion (Sec. 4.2) | 17.3 | 20.8 | 22.6 | 24.6 | 30.1 | 32.0 |
| + Reflectional matching (Sec. 4.3) | 19.0 | 22.4 | 23.8 | **26.4** | 30.5 | 32.2 |
| + Multi-scale matching (Sec. 4.4) | **19.7** | **23.9** | **25.7** | 26.2 | **31.0** | **33.6** |

true positive if it satisfies either of two conditions: 1) Endpoint condition: $d_1^2 + d_2^2 < \tau$, where $d_1$ and $d_2$ are the distances between the endpoints of the predicted line $l_p$ and the ground truth line $l_g$. 2) Ellipse condition: $d_{\text{center}}^2 < \tau$ with at least 70% overlap, where $d_{\text{center}}$ is the distance between the center of $l_p$ and the center of the ground truth ellipse.

The overlap is calculated as the percentage of $l_p$ that lies within the ellipse mask. We compute sAP at thresholds $\tau = 5, 10, 15$ pixels, denoted as sAP$^5$, sAP$^{10}$, and sAP$^{15}$ respectively. To prevent bias from images with many lines, we also evaluate an image-wise sAP: sAP$_{\text{img}}^\tau = \frac{1}{N} \sum \text{sAP}_i^\tau$ where $N$ is the number of images, and sAP$_i^\tau$ is the sAP for the $i$-th image. For comparison with segmentation-based benchmarks, we use the F1-score, defined as $\text{F1} = \frac{2 \times \text{precision} \times \text{recall}}{\text{precision} + \text{recall}}$. Following (Seo et al., 2022), both ground-truth and predicted score maps are dilated by 5 pixels (Funk & Liu, 2017) before computing true positives via pixel-wise comparison.

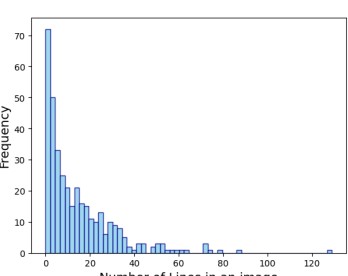

Figure 2: Line count distribution.

**Model and training.** We use the $D_8$-equivariant ResNet-34 He et al. (2016) as a feature extractor, keeping it frozen due to overfitting caused by the small amount of data while training. We freeze the backbone due to overfitting caused by the small amount of data while training. We maintain the same number of physical channel as the original network while achieving at least 16 times parameter efficiency. The reflectional matching module uses multi-receptive field processing (1, 3, and 5) with padded feature maps. In the branch network, equivariant deformable convolution is implemented by rotating the image, permuting the group dimension channels, applying standard deformable convolution Dai et al. (2017), and then rotating it back, since this operation is not natively supported by the e2cnn (Weiler & Cesa, 2019) framework. The model trains with a batch size of 64 for 100 epochs using the AdamW (Kingma & Ba, 2015) optimizer, starting with a learning rate of $1 \times 10^{-3}$, which reduces at the 50th and 75th epochs. The ground truth mid-point map represents each mid-point as a Gaussian distribution. The loss weights are set as $\lambda_\rho = 1$ for length, and $\lambda_\theta = 150$ for orientation to compensate for the smaller radian scale. Weighted binary cross-entropy with a positive class weight of 3 is used for mid-point detection.

## 5.2 Evaluation of Proposed Method

**Quantitative results.** Quantitative analysis of our key contributions is presented in Tab. 1. The table highlights the impact of accumulating our contributions to naive Axis-level detection (Sec. 4.1). Starting from axis-level detection alone, the addition of orientational anchor expansion significantly improves performance across all thresholds. This improvement demonstrates the importance of utilizing a specific part of the group equivariant feature map that responds to the direction a line is pointing. This approach is natural because the orientation response of the feature map directly corresponds to the line's direction. Incorporating reflectional matching boosts performance, indicating the benefits of group-aware matching that incorporating both spatial and group dimensions. Our final model, which incorporates multi-scale matching of the reflectional matching module, achieves the best results. This outcome demonstrates that capturing multiple receptive fields is crucial for detecting symmetry across various scales and transformations.

**Qualitative results.** The qualitative comparisons of the previous (Seo et al., 2021; 2022), and our method are shown in Fig. 3. In the ground truth and our images, symmetry axes are shown in green and ellipses in blue. Our method shows notable improvements over previous approaches. Un-

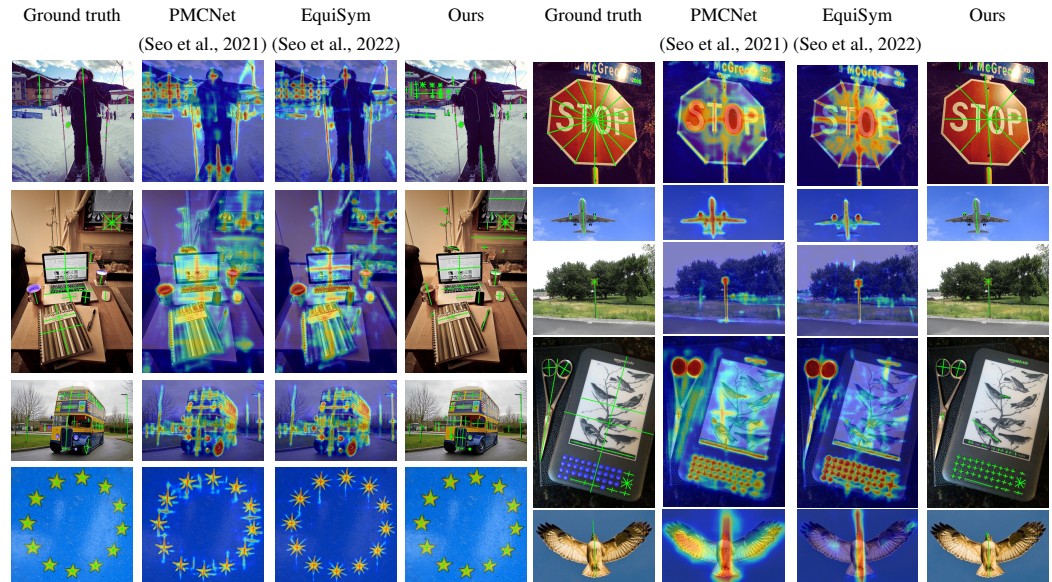

Figure 3: **Qualitative comparison of reflectional symmetry detection methods.** Our axis-wise approach produces clearer, more precise symmetry axes compared to pixel-wise methods (Seo et al., 2021; 2022), especially for smaller objects and complex scenes. Green lines in ground truth and our results represent symmetry axes, while blue regions indicate ellipses.

Table 2: Ablation on matching kernels in Reflectional Matching.

| Feature | Matching kernel | | | sAP | | | sAP$_{img}$ | | |
|---|---|---|---|---|---|---|---|---|---|
| | $1\times1$ | $3\times3$ | $5\times5$ | @5 | @10 | @15 | @5 | @10 | @15 |
| Base feature ($\mathbf{F}$) | - | - | - | 17.3 | 20.8 | 22.6 | 24.6 | 30.1 | 32.0 |
| RM feature ($\mathbf{H}$) | ✓ | | | 11.9 | 16.2 | 18.7 | 21.4 | 26.2 | 28.4 |
| | ✓ | ✓ | | 13.8 | 17.9 | 19.5 | **22.8** | 27.3 | 29.7 |
| | ✓ | ✓ | ✓ | **15.6** | **20.0** | 21.8 | 21.6 | **27.9** | **30.4** |
| [$\mathbf{F}$, $\mathbf{H}$] | ✓ | | | 19.0 | 22.4 | 23.8 | **26.4** | 30.5 | 32.2 |
| | ✓ | ✓ | | 18.5 | 22.4 | 23.7 | 25.2 | 30.3 | 32.6 |
| | ✓ | ✓ | ✓ | **19.7** | **23.9** | **25.7** | 26.2 | **31.0** | **33.6** |

like pixel-level detection methods (Seo et al., 2021; 2022) that focus on heat-map prediction, our axis-level detection produces clearer and more accurate results. Across different scenes and object types, our approach consistently provides more precise and interpretable symmetry axis predictions, especially in smaller objects where previous methods struggled with localization and orientation prediction. Our method generates sharper, well-defined axis predictions, making it suitable for analysis of precise reflecitonal symmetry.

## 5.3 ABLATION STUDY

**Matching kernels.** The analysis of different kernel configurations in the reflectional matching module reveals consistent performance improvements with larger receptive fields. Combining $1\times1$, $3\times3$, and $5\times5$ kernels leads to the highest sAP$^{10}$ of 23.9, compared to 22.4 with a $1\times1$ kernel alone (Tab. 2). This underscores the importance of spatial expansion for detecting various symmetry regions, confirming that multi-scale kernels are key for capturing more complex symmetry patterns. Notably, the performance gain from increasing kernel size is more pronounced when using reflectional matching alone compared to combining it with base features. This heightened sensitivity to multi-scale kernels in the reflectional matching module, which directly depends on these kernels, further emphasizes the effectiveness of this approach in capturing diverse symmetry patterns.

**Matching strategies.** A comparison of different matching strategies is shown in Tab. 3. This evaluation aims to assess the effectiveness of transformations involving the group dimension in matching

Table 3: Ablation on matching strategies.

| Method | sAP | | | sAP$_{\mathrm{img}}$ | | |
|---|---|---|---|---|---|---|
| | @5 | @10 | @15 | @5 | @10 | @15 |
| Spatial-only (w/o base) | 10.8 | 15.3 | 17.0 | 19.2 | 24.1 | 26.1 |
| Reflectional (w/o base) | **15.6** | **20.0** | **21.8** | **21.6** | **27.9** | **30.4** |
| Spatial-only | 18.3 | 21.8 | 22.9 | **26.3** | **31.8** | 33.5 |
| Reflectional | **19.7** | **23.9** | **25.7** | 26.2 | 31.0 | **33.6** |

Table 4: Pixel-level F1 score evaluation.

| Method | F1 score |
|---|---|
| PMCNet (Seo et al., 2021) | 32.6 |
| EquiSym (Seo et al., 2022) | 36.7 |
| Ours | **37.2** |

process, contrasting with spatial-only feature matching Seo et al. (2021). The reflectional matching score involves transforming feature pairs *w.r.t.* rotation angles in both spatial and group dimensions (Eq. (22)), while the spatial-only matching score uses spatial transformations alone, computed as $\mathbf{H}_{\mathbf{x}}^{(k)} = \sum_{\mathbf{q} \in \mathcal{Q}_k} \bigoplus_{n=0}^{|C_N|-1} h(\mathbf{F}_{\mathbf{x}+r^n(\mathbf{q})}, \mathbf{F}_{\mathbf{x}+br^n(\mathbf{q})})$. The results show consistent improvements with group-aware reflectional matching across both settings, demonstrating its effectiveness over spatial-only methods. The greater improvement without base features, as seen in kernel ablations, underscores the importance of reflectional matching in capturing symmetry axes.

**Image normalized sAP.** In addition to standard sAP Zhou et al. (2019), we also evaluate our model using sAP$_{\mathrm{img}}$ to mitigate potential bias from images with exceptionally large number of line annotations. Fig. 2 presents the distribution of the number of lines detected in test images, where the presence of images with an exceptionally high number of lines could skew the results. While sAP$_{\mathrm{img}}$ generally follows sAP trends, it may understate performance on complex scenes due to its image-wise normalization. However, the strong correlation between these metrics demonstrates our model's consistent effectiveness across varying scene complexities.

## 5.4 COMPARISON WITH THE STATE-OF-THE-ART METHODS

The results in Tab. 4 show a performance comparison for reflectional symmetry detection on the DENDI (Seo et al., 2022) *test* set using the F1-score for heatmap-based predictions. We exclude ellipse masks in this evaluation as they represent SO(2) continuous symmetry, which is expressed as a circular mask formed by an infinite number of symmetry lines. This is unsuitable for evaluating our method, which focuses on detecting discrete symmetry axes. Our method achieves an F1-score of 37.2, surpassing EquiSym (Seo et al., 2022)'s 36.7 and PMCNet (Seo et al., 2021)'s 32.6, demonstrating the superior ability of our approach in accurately localizing symmetry axes.

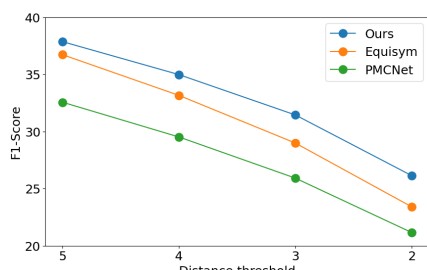

Figure 4: F1-score vs distance thresholds.

In Fig. 4, F1-scores for all three methods are plotted across different distance thresholds, with a true positive defined as within 5, 4, 3, or 2 pixels from ground-truth pixels. The plot highlights that our method achieves more precise localization than previous methods, particularly as the distance threshold tightens, demonstrating its superior accuracy.

## 6 CONCLUSION

We have introduced a novel axis-level reflectional symmetry detection network, leveraging dihedral group-equivariant representations to move beyond conventional pixel-level approaches. Our method incorporates several key contributions that significantly enhance performance across various challenging scenarios. First, we propose an orientational anchor expansion that enables fine-grained, rotation-equivariant analysis of symmetry patterns across multiple orientations. Second, we develop a reflectional matching module that uses multi-scale kernels to capture reflectional correlations, improving robustness across different receptive fields. Through extensive experiments, we demonstrate that our approach consistently outperforms existing methods, establishing a new benchmark in reflectional symmetry detection.

Looking ahead, expanding this framework to continuous groups and extending symmetry detection to 3D spaces, along with addressing viewpoint variations, represents promising directions for future work. These advancements could further broaden the applicability of symmetry detection in dynamic and real-world environments.

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

# A REGULAR REPRESENTATION AND GROUP CONVOLUTION

## A.1 DISCRETE GROUP REPRESENTATION

**Regular group representation.** The regular representation of a finite group $G = \{g_1, \ldots, g_N\}$ acts on a vector space $\mathbb{R}^{|G|}$. For any element $g \in G$, the regular representation $\sigma_{\text{reg}}^G(g)$ is defined as:

$$\sigma_{\text{reg}}^G(g) = [\mathbf{e}_{g \cdot g_1}, \ldots, \mathbf{e}_{g \cdot g_N}], \tag{23}$$

where each group element $g_i \in G$ is associated with a basis vector $\mathbf{e}_{g_i} \in \mathbb{R}^{|G|}$. In regular representation representation, $\sigma_{\text{reg}}^G(g) \in \mathbb{R}^{|G| \times |G|}$ is a permutation matrix that maps each basis vector $\mathbf{e}_{g_i}$ to $\mathbf{e}_{g \cdot g_i}$ for all $g_i \in G$.

**Cyclic group representation.** The cyclic group $\mathrm{C}_N$, consisting of $N$ discrete planar rotations, is defined as $\{r^0, r^1, \ldots, r^{(N-1)}\}$ with rotation generator $r$. With the group law $r^a \cdot r^b = r^{(a+b) \bmod N}$, the regular representation of $r^n$ is given by:

$$\sigma_{\text{reg}}^{\mathrm{C}_N}(r^n) = \left[\mathbf{e}_{r^n}, \mathbf{e}_{r^{(n+1) \bmod N}}, \ldots, \mathbf{e}_{r^{(n+N-1) \bmod N}}\right], \tag{24}$$

where the basis vectors are defined from:

$$\sigma_{\text{reg}}^{\mathrm{C}_N}(r^0) = \mathrm{I}_N, \tag{25}$$

where $\mathrm{I}_N$ being the $N \times N$ identity matrix. Here, the regular representation of the cyclic group corresponds to a cyclic permutation matrix.

**Dihedral group representation.** The dihedral group $\mathrm{D}_N = \{r^0, \ldots, r^{N-1}, b, rb, \ldots, r^{N-1}b\}$, consisting of $2N$ elements, is an extension of the cyclic group that includes an additional reflection generator $b$. The regular representation of the element $r^n b$ is given by:

$$\begin{aligned}
\sigma_{\text{reg}}^{\mathrm{D}_N}(r^n b) &= [\mathbf{e}_{r^n b}, \mathbf{e}_{r^n b \cdot r}, \ldots, \mathbf{e}_{r^n b \cdot r^{N-1}}, \mathbf{e}_{r^n b \cdot b}, \mathbf{e}_{r^n b \cdot rb}, \ldots, \mathbf{e}_{r^n b \cdot r^{N-1}b}] \\
&= [\mathbf{e}_{r^n b}, \mathbf{e}_{br^{n-1}}, \ldots, \mathbf{e}_{br^{n-N}}, \mathbf{e}_{r^n}, \mathbf{e}_{r^{n-1}}, \ldots, \mathbf{e}_{r^{n+1-N}}],
\end{aligned} \tag{26}$$

using the group laws $b^2 = e$ and $r^n b = br^{-n}$. By changing the order of cyclic rotation and reflection, the equation can be transformed as:

$$\begin{aligned}
\sigma_{\text{reg}}^{\mathrm{D}_N}(br^n) &= [\mathbf{e}_{br^n}, \mathbf{e}_{br^n \cdot r}, \ldots, \mathbf{e}_{br^n \cdot r^{N-1}}, \mathbf{e}_{br^n \cdot b}, \mathbf{e}_{br^n \cdot rb}, \ldots, \mathbf{e}_{br^n \cdot r^{N-1}b}] \\
&= [\mathbf{e}_{br^n}, \mathbf{e}_{br^{(n+1) \bmod N}}, \ldots, \mathbf{e}_{br^{(n+N-1) \bmod N}}, \mathbf{e}_{r^n}, \mathbf{e}_{r^{(n+1) \bmod N}}, \ldots, \mathbf{e}_{r^{(n+N-1) \bmod N}}].
\end{aligned} \tag{27}$$

The basis vectors for the dihedral group are defined from:

$$\sigma_{\text{reg}}^{\mathrm{D}_N}(r^0 b^0) = \mathrm{I}_{2N}. \tag{28}$$

## A.2 DISCRETE GROUP CONVOLUTION

Conventional convolutional neural networks (CNNs) are inherently equivariant to translations, meaning that a translation of the input results in a corresponding translation of the output. The standard 2D convolution operation can be expressed as:

$$(f * \psi)(\mathbf{x}) = \sum_{\mathbf{y} \in \mathbb{Z}^2} f(\mathbf{y}) \psi(\mathbf{x} - \mathbf{y}), \tag{29}$$

where $f : \mathbb{Z}^2 \to \mathbb{R}^{\mathcal{C}_{\text{in}}}$ is the input function with $\mathcal{C}_{\text{in}}$ channels, $\psi : \mathbb{Z}^2 \to \mathbb{R}^{\mathcal{C}_{\text{in}} \times \mathcal{C}_{\text{out}}}$ is the filter, and $\mathbf{x}, \mathbf{y} \in \mathbb{Z}^2$ are spatial coordinates. Here, plane feature map is defined only along the spatial dimension $\mathbb{Z}^2$. To associate discrete group within the feature map, an additional dimension corresponding to the group $G$ should be constructed, resulting in the mapping $f_G : G \times \mathbb{Z}^2 \to \mathbb{R}^{\mathcal{C}}$. In the discrete group convolution, this additional dimension is constructed through the lifting operation:

$$f_G = \bigoplus_{g \in G} (f * g\psi). \tag{30}$$

The order of the stack corresponds to the order of group elements in the initial state. Since the lifted feature map contains features corresponding to each group element, transformations must account for both spatial changes and the group structure. Applying a specific group element $g' \in G$ to the lifted feature map thus requires both spatial transformation and permutation of the group dimension:

$$(g' \cdot f_G)(\mathbf{x}) = \sigma_{\text{reg}}^G(g') \cdot f_G(g'^{-1}\mathbf{x}), \tag{31}$$

where $\sigma_{\text{reg}}^G(g')$ is the block diagonal form of the regular representation of $g'$ repeated $\mathcal{C}$ times, permuting along the group dimension, while $g'^{-1} \cdot \mathbf{x}$ applies the spatial transformation. Following the lifting operation, group convolution for the lifted feature map is defined as:

$$[f_G *_G \psi](g, \mathbf{x}) = \sum_{g' \in G} \sum_{\mathbf{y} \in \mathbb{Z}^2} f_G(g', \mathbf{y}) \left[\sigma_{\text{reg}}^G(g)\psi(g^{-1}(\mathbf{x} - \mathbf{y}))\right](g'). \tag{32}$$

Here, $\psi : G \times \mathbb{Z}^2 \to \mathbb{R}^{\mathcal{C}_{\text{in}} \times \mathcal{C}_{\text{out}}}$ represents the group convolution filter, where $f_G : G \times \mathbb{Z}^2 \to \mathbb{R}^{\mathcal{C}_{\text{in}}}$ is the lifted feature map, and $g, g' \in G$ are group elements of $G$. The key property of group convolution is its equivariance to group elements, expressed as:

$$[(g' \cdot f_G) *_G \psi](\mathbf{x}) = [g' \cdot (f_G *_G \psi)](\mathbf{x})$$
$$= \sigma_{\text{reg}}^G(g') \cdot (f_G *_G \psi)(g'^{-1} \cdot \mathbf{x}) \tag{33}$$

for any $g' \in G$. Here, $(g' \cdot f_G) *_G \psi$ represents the group convolution applied to the transformed input, while $g' \cdot (f_G *_G \psi)$ is the action of $g'$ on the result of the group convolution. This equality demonstrates that the order of applying group transformations and group convolutions is interchangeable, preserving the group structure throughout the network layers.

## B $\quad$ C$_N$ EQUIVARIANCE OF THE REFLECTIONAL MATCHING

### B.1 $\quad$ C$_N$ EQUIVARIANCE OF THE SINGLE FIBER REFLECTIONAL MATCHING

Given a D$_N$-equivariant feature map $\mathbf{F} \in \mathbb{R}^{\mathcal{C}|\text{D}_N| \times H \times W}$ under the regular representation $\sigma_{\text{reg}}$, we need to prove that $\mathbf{H}$ from Reflectional Matching without spatial expansion is equivariant to the cyclic group C$_N$ with its element $r^k$:

$$\bigoplus_{n=0}^{N-1} h\left(\sigma_{\text{reg}}^{\text{D}_N}(r^n)\mathbf{F}_{\mathbf{x}}^{(0,k)}, \sigma_{\text{reg}}^{\text{D}_N}(br^n)\mathbf{F}_{\mathbf{x}}^{(0,k)}\right) \tag{34}$$

$$= \sigma_{\text{reg}}^{\text{D}_N}(r^k) \bigoplus_{n=0}^{N-1} h\left(\sigma_{\text{reg}}^{\text{D}_N}(r^n)\mathbf{F}_{\mathbf{x}}^{(0,0)}, \sigma_{\text{reg}}^{\text{D}_N}(br^n)\mathbf{F}_{\mathbf{x}}^{(0,0)}\right), \tag{35}$$

where $\mathbf{F}_{\mathbf{x}}^{(l,n)}$ is the fiber at position $\mathbf{x}$, with the regular representation corresponding to $l$ reflections and $n$ rotations added. Using the property $\sigma(g)\sigma(h) = \sigma(gh)$, the equation can be rewritten as:

$$\bigoplus_{n=0}^{N-1} h\left(\sigma_{\text{reg}}^{\text{D}_N}(r^n)\mathbf{F}_{\mathbf{x}}^{(0,k)}, \sigma_{\text{reg}}^{\text{D}_N}(br^n)\mathbf{F}_{\mathbf{x}}^{(0,k)}\right) \tag{36}$$

$$= \bigoplus_{n=0}^{N-1} h\left(\sigma_{\text{reg}}^{\text{D}_N}(r^n)\sigma_{\text{reg}}^{\text{D}_N}(r^k)\mathbf{F}_{\mathbf{x}}^{(0,0)}, \sigma_{\text{reg}}^{\text{D}_N}(br^n)\sigma_{\text{reg}}^{\text{D}_N}(r^k)\mathbf{F}_{\mathbf{x}}^{(0,0)}\right) \tag{37}$$

$$= \bigoplus_{n=0}^{N-1} h\left(\sigma_{\text{reg}}^{\text{D}_N}(r^{k+n})\mathbf{F}_{\mathbf{x}}^{(0,0)}, \sigma_{\text{reg}}^{\text{D}_N}(br^{k+n})\mathbf{F}_{\mathbf{x}}^{(0,0)}\right). \tag{38}$$

Here, $h$ is the similarity function defined as:

$$h(\mathbf{f}^1, \mathbf{f}^2) = \bigoplus_{c=1}^{\mathcal{C}} \frac{\mathbf{f}_c^1 \cdot \mathbf{f}_c^2}{\|\mathbf{f}_c^1\|\|\mathbf{f}_c^2\|} \in \mathbb{R}^{\mathcal{C}}, \tag{39}$$

Since permutation matrices preserve the norm of a vector, and using the rule $r^{a+b} = r^{(a+b) \bmod N}$, the equation can be reformulated as:

$$\bigoplus_{n=0}^{N-1} \bigoplus_{c=1}^{\mathcal{C}} \frac{1}{\|\mathbf{F}_{c,\mathbf{x}}^{(0,0)}\|^2} \left( \sigma_{\text{reg}}^{\text{D}_N}(r^{k+n}) \mathbf{F}_{c,\mathbf{x}}^{(0,0)} \cdot \sigma_{\text{reg}}^{\text{D}_N}(br^{k+n}) \mathbf{F}_{c,\mathbf{x}}^{(0,0)} \right) \tag{40}$$

$$= \bigoplus_{c=1}^{\mathcal{C}} \frac{1}{\|\mathbf{F}_{c,\mathbf{x}}^{(0,0)}\|^2} \bigoplus_{n=0}^{N-1} \left( \sigma_{\text{reg}}^{\text{D}_N}(r^{k+n}) \mathbf{F}_{c,\mathbf{x}}^{(0,0)} \cdot \sigma_{\text{reg}}^{\text{D}_N}(br^{k+n}) \mathbf{F}_{c,\mathbf{x}}^{(0,0)} \right) \tag{41}$$

$$= \bigoplus_{c=1}^{\mathcal{C}} \frac{1}{\|\mathbf{F}_{c,\mathbf{x}}^{(0,0)}\|^2} \bigoplus_{n=k}^{k+N-1} \left( \sigma_{\text{reg}}^{\text{D}_N}(r^{n}) \mathbf{F}_{c,\mathbf{x}}^{(0,0)} \cdot \sigma_{\text{reg}}^{\text{D}_N}(br^{n}) \mathbf{F}_{c,\mathbf{x}}^{(0,0)} \right) \tag{42}$$

$$= \bigoplus_{c=1}^{\mathcal{C}} \frac{1}{\|\mathbf{F}_{c,\mathbf{x}}^{(0,0)}\|^2} \sigma_{\text{reg}}^{\text{D}_N}(r^{k}) \bigoplus_{n=0}^{N-1} \left( \sigma_{\text{reg}}^{\text{D}_N}(r^{n}) \mathbf{F}_{c,\mathbf{x}}^{(0,0)} \cdot \sigma_{\text{reg}}^{\text{D}_N}(br^{n}) \mathbf{F}_{c,\mathbf{x}}^{(0,0)} \right) \tag{43}$$

$$= \sigma_{\text{reg}}^{\text{D}_N}(r^{k}) \bigoplus_{c=1}^{\mathcal{C}} \frac{1}{\|\mathbf{F}_{c,\mathbf{x}}^{(0,0)}\|^2} \bigoplus_{n=0}^{N-1} \left( \sigma_{\text{reg}}^{\text{D}_N}(r^{n}) \mathbf{F}_{c,\mathbf{x}}^{(0,0)} \cdot \sigma_{\text{reg}}^{\text{D}_N}(br^{n}) \mathbf{F}_{c,\mathbf{x}}^{(0,0)} \right) \tag{44}$$

$$= \sigma_{\text{reg}}^{\text{D}_N}(r^{k}) \bigoplus_{n=0}^{N-1} h\left( \sigma_{\text{reg}}^{\text{D}_N}(r^{n}) \mathbf{F}_{\mathbf{x}}^{(0,0)}, \sigma_{\text{reg}}^{\text{D}_N}(br^{n}) \mathbf{F}_{\mathbf{x}}^{(0,0)} \right), \tag{45}$$

where $\mathbf{F}_{c,\mathbf{x}}$ denotes the feature at position $\mathbf{x}$ in channel $c$.

## B.2 $C_N$ Equivariance of Spatially Expanded Reflectional Matching

We now have to to prove the spatial expansion of single fiber Reflectional Matching is also equivariant to the cyclic group $C_N$:

$$\bigoplus_{n=0}^{N-1} \sum_{\mathbf{q} \in \mathcal{Q}} h\left( \sigma_{\text{reg}}^{\text{D}_N}(r^{n}) \mathbf{F}_{\mathbf{x}+r^{k+n}(\mathbf{q})}^{(0,k)}, \sigma_{\text{reg}}^{\text{D}_N}(br^{n}) \mathbf{F}_{\mathbf{x}+br^{k+n}(\mathbf{q})}^{(0,k)} \right) \tag{46}$$

$$= \sigma_{\text{reg}}^{\text{D}_N}(r^{k}) \bigoplus_{n=0}^{N-1} \sum_{\mathbf{q} \in \mathcal{Q}} h\left( \sigma_{\text{reg}}^{\text{D}_N}(r^{n}) \mathbf{F}_{\mathbf{x}+r^{n}(\mathbf{q})}^{(0,0)}, \sigma_{\text{reg}}^{\text{D}_N}(br^{n}) \mathbf{F}_{\mathbf{x}+br^{n}(\mathbf{q})}^{(0,0)} \right), \tag{47}$$

where $\mathbf{q} \in \mathcal{Q}$ is the offset, $r^{n}(\mathbf{q})$ represents the spatially rotated offset, and $br^{n}(\mathbf{q})$ denotes the offset that is first rotated and then reflected. Same as single fiber, the equation can be written as:

$$\bigoplus_{n=0}^{N-1} \sum_{\mathbf{q} \in \mathcal{Q}} h\left( \sigma_{\text{reg}}^{\text{D}_N}(r^{k+n}) \mathbf{F}_{\mathbf{x}+r^{k+n}(\mathbf{q})}^{(0,0)}, \sigma_{\text{reg}}^{\text{D}_N}(br^{k+n}) \mathbf{F}_{\mathbf{x}+br^{k+n}(\mathbf{q})}^{(0,0)} \right) \tag{48}$$

$$= \bigoplus_{n=0}^{N-1} \sum_{\mathbf{q} \in \mathcal{Q}} \bigoplus_{c=1}^{\mathcal{C}} \frac{\sigma_{\text{reg}}^{\text{D}_N}(r^{k+n}) \mathbf{F}_{c,\mathbf{x}+r^{k+n}(\mathbf{q})}^{(0,0)} \cdot \sigma_{\text{reg}}^{\text{D}_N}(br^{k+n}) \mathbf{F}_{c,\mathbf{x}+br^{k+n}(\mathbf{q})}^{(0,0)}}{\|\mathbf{F}_{c,\mathbf{x}+r^{k+n}(\mathbf{q})}^{(0,0)}\| \|\mathbf{F}_{c,\mathbf{x}+br^{k+n}(\mathbf{q})}^{(0,0)}\|} \tag{49}$$

$$= \bigoplus_{c=1}^{\mathcal{C}} \bigoplus_{n=k}^{k+N-1} \sum_{\mathbf{q} \in \mathcal{Q}} \frac{\sigma_{\text{reg}}^{\text{D}_N}(r^{n}) \mathbf{F}_{c,\mathbf{x}+r^{n}(\mathbf{q})}^{(0,0)} \cdot \sigma_{\text{reg}}^{\text{D}_N}(br^{n}) \mathbf{F}_{c,\mathbf{x}+br^{n}(\mathbf{q})}^{(0,0)}}{\|\mathbf{F}_{c,\mathbf{x}+r^{n}(\mathbf{q})}^{(0,0)}\| \|\mathbf{F}_{c,\mathbf{x}+br^{n}(\mathbf{q})}^{(0,0)}\|} \tag{50}$$

$$= \bigoplus_{c=1}^{\mathcal{C}} \sigma_{\text{reg}}^{\text{D}_N}(r^{k}) \bigoplus_{n=0}^{N-1} \sum_{\mathbf{q} \in \mathcal{Q}} \frac{\sigma_{\text{reg}}^{\text{D}_N}(r^{n}) \mathbf{F}_{c,\mathbf{x}+r^{n}(\mathbf{q})}^{(0,0)} \cdot \sigma_{\text{reg}}^{\text{D}_N}(br^{n}) \mathbf{F}_{c,\mathbf{x}+br^{n}(\mathbf{q})}^{(0,0)}}{\|\mathbf{F}_{c,\mathbf{x}+r^{n}(\mathbf{q})}^{(0,0)}\| \|\mathbf{F}_{c,\mathbf{x}+br^{n}(\mathbf{q})}^{(0,0)}\|} \tag{51}$$

$$= \sigma_{\text{reg}}^{\text{D}_N}(r^{k}) \bigoplus_{n=0}^{N-1} \sum_{\mathbf{q} \in \mathcal{Q}} \bigoplus_{c=1}^{\mathcal{C}} \frac{\sigma_{\text{reg}}^{\text{D}_N}(r^{n}) \mathbf{F}_{c,\mathbf{x}+r^{n}(\mathbf{q})}^{(0,0)} \cdot \sigma_{\text{reg}}^{\text{D}_N}(br^{n}) \mathbf{F}_{c,\mathbf{x}+br^{n}(\mathbf{q})}^{(0,0)}}{\|\mathbf{F}_{c,\mathbf{x}+r^{n}(\mathbf{q})}^{(0,0)}\| \|\mathbf{F}_{c,\mathbf{x}+br^{n}(\mathbf{q})}^{(0,0)}\|} \tag{52}$$

$$= \sigma_{\text{reg}}^{\text{D}_N}(r^{k}) \bigoplus_{n=0}^{N-1} \sum_{\mathbf{q} \in \mathcal{Q}} h\left( \sigma_{\text{reg}}^{\text{D}_N}(r^{n}) \mathbf{F}_{\mathbf{x}+r^{n}(\mathbf{q})}^{(0,0)}, \sigma_{\text{reg}}^{\text{D}_N}(br^{n}) \mathbf{F}_{\mathbf{x}+br^{n}(\mathbf{q})}^{(0,0)} \right), \tag{53}$$

## C   ADDITIONAL EXPERIMENTS AND ANALYSIS

### C.1   EVALUATION ON DIFFERENT DATASETS

We conduct experiments on the LDRS (Liu et al., 2013) and SDRW (Seo et al., 2022) datasets to demonstrate the generalizability and applicability of our approach across different datasets.

Table 5: Quantitative comparison of F1-scores for DENDI (Seo et al., 2022), LDRS (Seo et al., 2021) and SDRW (Liu et al., 2013) datasets.

| Method | DENDI | LDRS | SDRW |
|---|---|---|---|
| PMCNet (Seo et al., 2021) | 32.6 | 37.3 | **68.8** |
| EquiSym (Seo et al., 2022) | 36.7 | 40.0 | 67.5 |
| Ours | **37.2** | **43.4** | 68.3 |

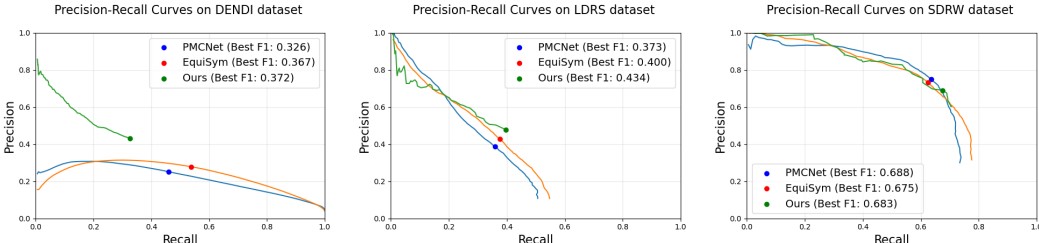

Figure 5: Precision-Recall curves and F1-scores evaluated on three different datasets (DENDI (Seo et al., 2022), LDRS (Seo et al., 2021), and SDRW (Liu et al., 2013)).

Tab. 5 shows that our method achieves state-of-the-art performance with an F1-score of 0.434 on the LDRS (Seo et al., 2021) dataset, significantly outperforming previous methods including PM-CNet (Seo et al., 2021) (0.373) and EquiSym (Seo et al., 2022) (0.400). For the SDRW (Liu et al., 2013) dataset, we achieve competitive results with an F1-score of 0.683, closely matching the best performance of PMCNet (0.686). As shown in Fig. 5, the precision-recall curves across different datasets demonstrate our method's characteristics. Notably, our approach exhibits higher precision but relatively lower recall compared to existing methods. This behavior stems from our post-processing pipeline, which includes Non-Maximum Suppression (Neubeck & Van Gool, 2006) and score-based thresholding on detected lines. In contrast, conventional pixel-level prediction methods tend to achieve higher recall at lower thresholds by treating all pixels as potential predictions.

### C.2   COMPARISON WITH PREVIOUS METHODS

To provide a comprehensive evaluation against existing approaches, we compare our method with a wide range of previous methods on the SDRW (Liu et al., 2013) dataset. Beyond recent approaches like PMCNet (Seo et al., 2021), we include comparisons with classical and modern approaches: SymResNet (Funk & Liu, 2017), LE (Loy & Eklundh, 2006), MIL (Tsogkas & Kokkinos, 2012), FSDS (Shen et al., 2016), and SRF (Teo et al., 2015). As shown in Fig. 6, our precision-recall curve demonstrates competitive performance, achieving an F1-score of 0.683, which is comparable to the state-of-the-art result achieved by PMCNet (0.686). This comparison with both recent and classical methods highlights the effectiveness of our approach within the context of symmetry detection.

### C.3   ABLATION STUDIES WITH DIFFERENT BACKBONES

To investigate the effectiveness of our approach across different backbone architectures, we conduct experiments with non-equivariant networks including ResNet-34 (He et al., 2016) and the more recent ConvNeXt (Liu et al., 2022). Due to the limited size of the DENDI (Seo et al., 2022) dataset and potential overfitting issues observed in our preliminary experiments, we maintain consistent experimental settings with our main results by using ImageNet (Deng et al., 2009) pre-trained encoders in a frozen state. For these non-equivariant architectures which lack the group dimension

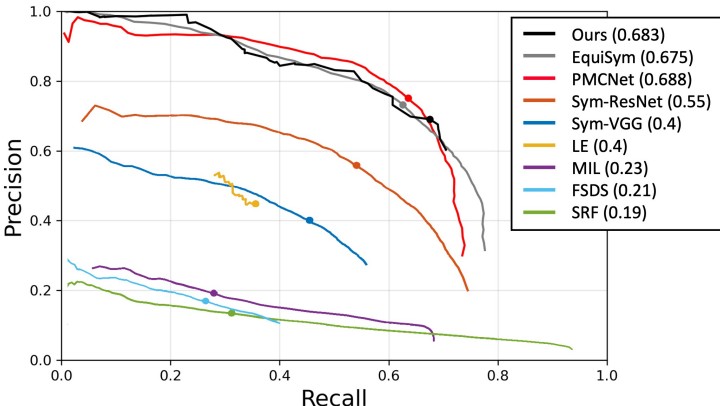

Figure 6: PR curve and F1-score comparison with previous methods on SDRW Liu et al. (2013) dataset.

structure, we adapt our method by implementing spatial matching instead of group-aware matching while maintaining the same orientational anchor setup (four anchors) with each anchor assigned a different ground truth orientation. Tab. 6 presents the comparative results across different backbone architectures and matching strategies.

Table 6: Performance comparison of detection architectures with varying matching and anchor configurations.

| Backbone | Method | sAP | | |
| | | @5 | @10 | @15 |
|---|---|---|---|---|
| ResNet-34 He et al. (2016) | Single Anchor | 4.7 | 7.9 | 10.1 |
| ResNet-34 He et al. (2016) | Orientational Anchor | 15.6 | 19.2 | 20.1 |
| ConvNeXt Liu et al. (2022) | Single Anchor | 11.1 | 15.5 | 17.3 |
| ConvNeXt Liu et al. (2022) | Orientational Anchor | 18.1 | 21.1 | 22.2 |
| Ours( Cohen & Welling (2016a)) | Orientational Anchor | **19.7** | **23.9** | **25.7** |

The results highlight several key findings. The use of orientational anchors consistently improves performance across all architectures, with significant gains observed in both ResNet-34 (He et al., 2016) (+10.9 in sAP@5) and ConvNeXt (Liu et al., 2022) (+7.0 in sAP@5). While ConvNeXt shows stronger performance compared to ResNet-34, particularly in the single anchor setting, both architectures still fall short of our group-aware approach when using orientational anchors. The superior performance of our group-aware approach (19.7/23.9/25.7) over non-equivariant architectures suggests that the combination of group-aware matching and orientational anchors aligned with the group dimension provides advantages that exceed those of using more modern architectures. These findings further validate our design choices and demonstrate that the equivariance properties and group-aware design of our approach are crucial to its effectiveness.

## C.4 ABLATION STUDIES ON LINE SEGMENT DETECTION

To validate the generalization capability of our method beyond symmetry detection, we conduct experiments on the line segment detection task using the Wireframe dataset (Huang et al., 2018), a standard benchmark for this task. Tab. 7 presents the comparative results with recent state-of-the-art methods. Our method achieves competitive performance compared to specialized line detection approaches, even without eliminating the offset and centerness branches from the original architecture (Zhang et al., 2021). These results demonstrate that our adaptation strategy effectively transfers to general line detection tasks while maintaining strong performance.

Table 7: Performance comparison with state-of-the-art methods on the Wireframe dataset.

| Method | sAP | | |
| --- | --- | --- | --- |
| | @5 | @10 | @15 |
| L-CNN Zhou et al. (2019) | 58.9 | 62.9 | 64.7 |
| HAWP Xu et al. (2021) | 62.5 | 66.5 | 68.2 |
| F-Clip Dai et al. (2022) | 64.3 | 68.3 | - |
| ELSD Zhang et al. (2021) | **64.3** | **68.9** | **70.9** |
| ELSD (reproduced) | 63.7 | 68.0 | 69.3 |
| Ours | 62.2 | 66.5 | 68.3 |

## C.5 ABLATION STUDIES ON COMPUTATIONAL OVERHEAD

We provide a detailed analysis of the computational requirements and performance trade-offs for different multi-scale configurations of our reflectional matching approach. Tab. 8 presents the computational costs and corresponding performance metrics.

Table 8: Computational overhead analysis for different features

| Feature | Matching kernel | | | Time(h) | Memory(GB) Per/Total | GFLOPs | sAP | | |
| --- | --- | --- | --- | --- | --- | --- | --- | --- | --- |
| | 1×1 | 3×3 | 5×5 | | | | @5 | @10 | @15 |
| **F** | - | - | - | 9.5 | 14 / 56 | 39.5 | 17.3 | 20.8 | 22.6 |
| **[F, H]** | ✓ | | | 13.0 | 22 / 88 | 58.5 | 19.0 | 22.4 | 23.8 |
| | ✓ | ✓ | | 14.5 | 26 / 104 | 81.2 | 18.5 | 22.4 | 23.7 |
| | ✓ | ✓ | ✓ | 17.0 | 30 / 120 | 102.5 | 19.7 | 23.9 | 25.7 |

As shown in Tab. 8, introducing multi-scale reflectional matching significantly increases computational requirements. Training on four RTX6000ADA GPUs, the base feature (**F**) requires 39.5 GFLOPs and 56G total GPU memory with a training time of 9.5 hours. Adding larger matching kernels progressively increases these requirements, with the full three-scale configuration requiring 102.5 GFLOPs, 120G total GPU memory, and 17 hours of training time. This computational overhead is compensated by performance improvements across evaluation metrics. Using all three matching kernels achieves the best results, showing improvements of +2.4, +3.1, and +3.1 points in sAP@5, @10, and @15 respectively compared to using base feature alone. These results suggest that our method is not yet fully optimized, and further improvements in the efficiency of the matching process could make it even more effective.

## D ARCHITECTURE DETAILS: MULTI-SCALE MATCHING AND ORIENTATIONAL AXIS RECONSTRUCTION

In Fig. 7, we present the detailed architecture for orientational axis reconstruction and multi-scale matching with the $D_8$ group. The feature maps **H** are obtained by applying reflectional matching modules with different offsets ($\mathcal{Q}$) to the base feature **F**Base, exhibiting cyclic group equivariance and reflection invariance (a. Multi-scale Matching). We then concatenate **H** from different offsets and combine them with the base feature $\mathbf{F}_{Base}$. Subsequently, this concatenated feature is transformed into a single-channel feature map with orientational dimensions through the cyclic group branch. Since orientation responses from $\theta$ and $\theta + \pi$ should be identical from a line perspective, we add the first four channels with the last four channels of a single-channel feature map. Each of the resulting four channels reconstructs lines using midpoint, length, and orientation parameters, where each anchor captures lines within a directional range of $\left[-\frac{\pi}{8}, \frac{\pi}{8}\right)$ from its corresponding orientation. Finally, these reconstructed axes are concatenated into a single-channel output (b. Orientational Axis Reconstruction).

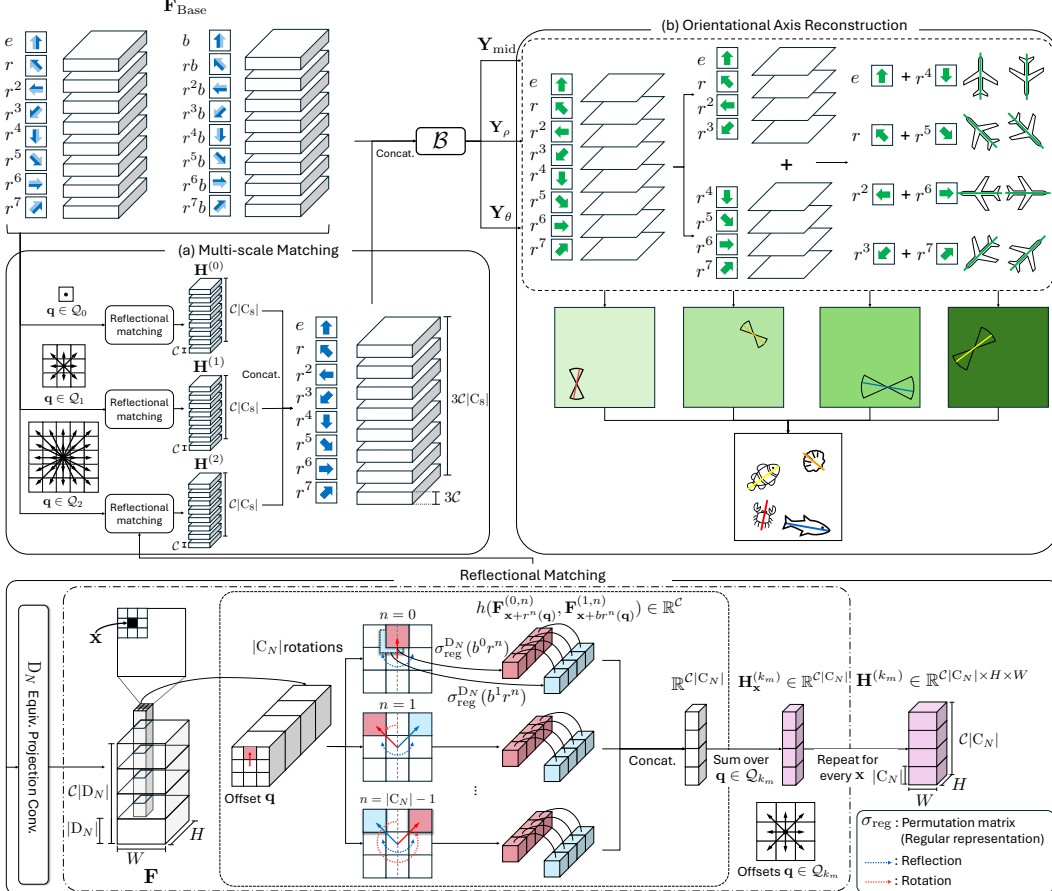

Figure 7: Multi-scale matching and orientational anchor expansion using the $D_8$ group. Blue and green arrows indicate filter orientations for each orientation channel. The four anchors (in green background) indicate the directional ranges for lines captured by each anchor, where each anchor detects lines aligned with its corresponding orientation. $\mathcal{B}$ and $\mathbf{Y}$ denote the branch and branch output from Eq. (14) respectively, while $\mathbf{H}$ represents the reflectional matching feature from Eq. (22). The reflectional matching component from Fig. 1 is also included.

