# OpenReview forum: "Axis-level Reflectional Symmetry Detection with Group-Equivariant Representation"
_ICLR.cc/2025/Conference — Submitted to ICLR 2025_

### Official Review · Reviewer_YGvV · 2024-11-03

**Soundness:** 3
**Presentation:** 3
**Contribution:** 3
**Rating:** 5
**Confidence:** 2

**Summary:**

The paper proposes a group-equivariant neural network for axis-level reflectional symmetry detection。The authors introduce orientational anchor expansion for fine-grained rotational equivariant analysis of different symmetry patterns across multiple orientations. Additionally, the paper develops reflectional matching with multi-scale kernels, enabling robust symmetry detection across various receptive fields. Experimental results demonstrate the effectiveness of the proposed method.

**Strengths:**

1.	The idea is interesting. Compared to existing methods, which primarily treat reflectional symmetry detection as a pixel-level heatmap prediction problem, this paper classifies the presence of a mid-point of a reflectional symmetry axis for each pixel position and also regress the angle and length of the axis , directly performing axis-level prediction.

2.	Extensive experiments validate the effectiveness of the proposed method, providing more accurate axis-level predictions than existing pixel-level methods.

3.	The paper is well-organized, making it easy and quick to follow.

**Weaknesses:**

1.	The literature review is incomplete. The cited references are almost entirely from 2022 and earlier (with only one paper from 2023 and none from 2024), raising questions about the novelty of the work.

2.	The proposed multi-scale expansion has already been widely explored and proven effective in tasks such as object detection and segmentation.

**Questions:**

1.	In Line 521, you claim that “In Fig. 4, F1-scores for all three methods are plotted across different distance thresholds,” but Fig. 4 only shows two methods, missing the presentation of PMCNet.

2.	The proposed method adapts a line detection network and applies it to the reflectional symmetry detection task. Can the adaptation strategy be effective on other line detection networks?

3.	Could you provide comparison results with existing methods on other datasets (such as SDRW[1] and LDRS[2]) to fully demonstrate the superiority of the proposed method?

4.	What are the application scenarios, research value, and significance of this study?

[1] Liu, Jingchen, et al. "Symmetry detection from realworld images competition 2013: Summary and results."
[2] Seo, Ahyun, Woohyeon Shim, and Minsu Cho. "Learning to discover reflection symmetry via polar matching convolution."
Flag For Ethics Review: No ethics review needed.

---

> ### Author Response · Authors · 2024-11-20
> **Response to reviewer YGvV(1/2)**
>
> >**W1:**
> >The literature review is incomplete. The cited references are almost entirely from 2022 and earlier (with only one paper from 2023 and none from 2024), raising questions about the novelty of the work.
>
> We appreciate your concern about the timeliness of our citations. In our literature review, we found that recent work (2023-2024) has primarily focused on 3D reflection symmetry (line 48) and domain-specific reflection symmetry applications, which we have cited accordingly (line 49).
>
> While papers explicitly addressing 2D reflection symmetry detection in a general context appear to be relatively scarce in recent literature, we would be genuinely grateful if you could bring to our attention any relevant works we might have missed. This would help us better position our work within the current research landscape and enhance our literature review.
>
> >**W2:**
> >The proposed multi-scale expansion has already been widely explored and proven effective in tasks such as object detection and segmentation.
>
> We appreciate the reviewer's insights regarding the terminology. Indeed, "multi-scale expansion" has been widely used across different works with varying contexts - it could refer to input scaling, feature map modifications, or kernel size variations. We acknowledge that our use of this term may not adequately convey the specific contribution of our method.
> To clarify our approach more precisely, the key aspect of our method in Sec. 4.4 is the incorporation of spatial matching into the reflectional matching framework described in Sec. 4.3. Specifically:
>
> In Equation 21, the reflectional matching focuses solely on group representation, where matching occurs between a feature and its rotated counterparts.
> This is extended in Equation 23 by incorporating a spatially-aware matching mechanism that rotates corresponding to the angle, enabling the matching process to consider multiple spatial locations beyond a single position.
>
> Beyond simply using multiple kernel sizes (1×1 as in Sec. 4.3, along with 3×3 and 5×5), our approach introduces the advantage of capturing both rotational and spatial correspondences in the matching process in multi-scales. We apologize for any confusion caused by our terminology and have revised it to "multi-scale matching"(Table 1, subsection 4.4, and line 426) to better reflect the distinct nature of our contribution.
>
> >**Q1:**
> >In Line 521, you claim that "In Fig. 4, F1-scores for all three methods are plotted across different distance thresholds," but Fig. 4 only shows two methods, missing the presentation of PMCNet.
>
> Thank you for catching this error. We acknowledge the reviewer's observation regarding the mismatch between the text and Fig. 4. We have revised the figure to accurately reflect all three methods described in the text. We apologize for any confusion this may have caused.
>
> >**Q2:**
> >The proposed method adapts a line detection network and applies it to the reflectional symmetry detection task. Can the adaptation strategy be effective on other line detection networks?
>
> We appreciate the question about application to the line segment detection task. Our method is an adaptation of ELSD[1] with modifications in the overall network architecture to make the network work better on the reflectional symmetry detection task. Although we discarded the offset and centerness branches of ELSD[1] (2 of the total 4 branches), our approach generalizes effectively to other line detection tasks.
> We have validated this by testing our method on the Wireframe dataset [2], which is a widely-used benchmark for line detection, achieving strong performance with sAP scores of 62.2/66.5/68.3 (at 5/10/15 thresholds). These results are comparable to our reproduced ELSD [1] (63.7/68.0/69.3) and close to their reported performance (64.3/68.9/70.9). This demonstrates that our method maintains high performance on general line detection even with fewer prediction branches, highlighting the effectiveness of our architectural modifications. The quantitative results can be found in our supplementary materials(C.4, Table 7).
>
> | Method | sAP@5 | sAP@10 | sAP@15 |
> |--------|-------|--------|---------|
> | L-CNN | 58.9 | 62.9 | 64.7 |
> | HAWP | 62.5 | 66.5 | 68.2 |
> | F-Clip | 64.3 | 68.3 | - |
> | ELSD | **64.3** | **68.9** | **70.9** |
> | ELSD (reproduced) | 63.7 | 68.0 | 69.3 |
> | Ours | 62.2 | 66.5 | 68.3 |
>
>
> **References:**
> 1. Zhang, Haotian, et al. "Elsd: Efficient line segment detector and descriptor." Proceedings of the IEEE/CVF International Conference on Computer Vision. 2021.
> 2. Huang, Kun, et al. "Learning to parse wireframes in images of man-made environments." Proceedings of the IEEE Conference on Computer Vision and Pattern Recognition. 2018.

---

> ### Author Response · Authors · 2024-11-20
> **Response to reviewer YGvV(2/2)**
>
> >**Q3:**
> >Could you provide comparison results with existing methods on other datasets (such as SDRW[1] and LDRS[2]) to fully demonstrate the superiority of the proposed method?
>
> We appreciate the reviewer’s suggestion to provide comparisons on additional datasets to fully demonstrate the effectiveness of our method. In response, we have expanded our evaluation and included results on the following datasets:
>
> **SDRW Dataset:**
> Our method achieved the second-best performance with an F1-score of 0.683, narrowly trailing PMCNet (Seo et. al. 2021), which achieved 0.686.
> This dataset includes comparisons with other prior methods such as PMCNet, SymResNet, LE, MIL, FSDS, and SRF.
>
> **LDRS Dataset:**
> Our method achieved state-of-the-art performance with an F1-score of 0.434, surpassing heat-map-based methods such as PMCNet (Seo et. al. 2021)(0.373) and EquiSym(Seo et. al. 2022)(0.400, reproduced).
>
> | Method | DENDI | LDRS | SDRW |
> |--------|-------|------|------|
> | PMCNet | 32.6 | 37.3 | **68.8** |
> | EquiSym | 36.7 | 40.0 | 67.5 |
> | Ours | **37.2** | **43.4** | 68.3 |
>
>
> These results highlight the strengths of our method in localization and detection schemes. Unlike heat-map-based methods, which often sacrifice precision for higher recall, our approach achieves competitive F1-scores by balancing precision and recall effectively. The results and corresponding precision-recall curves are included in the supplemental material for detailed comparison.
>
> >**Q4:**
> >What are the application scenarios, research value, and significance of this study?
>
> This study has substantial application potential, research value, and significance across various domains.
>
> **Application Scenarios**
> Reflection symmetry is widely used across various domains to improve efficiency and accuracy. It helps in real-time object tracking [1,2], 3D reconstruction [3], and 6D pose estimation [4]. In face recognition, it enhances classification by analyzing facial asymmetry [5], and in medical imaging, it aids in detecting and segmenting brain tumors [6]. Symmetry also improves text recognition through pattern rectification [7], strengthens human tracking and gait recognition [2], and ensures structural balance in image retargeting for various aspect ratios [8].
>
> **Research Value**
> This study bridges low-level image features and high-level tasks by advancing computational symmetry in vision, particularly for curved and reflection symmetries. Its contributions address challenges across medical imaging, robotics, and computer graphics, highlighting its interdisciplinary impact.
>
> **Significance**
> The study introduces a novel reflection symmetry detection network using dihedral group-equivariant representation, orientational anchor expansion, and multi-scale reflectional matching. These components achieve robust symmetry detection, validated through superior performance in real-world scenarios.
>
> **References**
> 1. Li, Wai Ho, Alan M. Zhang, and Lindsay Kleeman. "Bilateral symmetry detection for real-time robotics applications." The International Journal of Robotics Research 27.7 (2008): 785-814.
> 2. W.H. Li and L. Kleeman. "Real-Time Object Tracking Using Reflectional Symmetry and Motion." Proc. IEEE Int’l Conf. Intelligent Robots and Systems, 2006.
> 3. Wu, Shangzhe, Christian Rupprecht, and Andrea Vedaldi. "Unsupervised learning of probably symmetric deformable 3D objects from images in the wild." CVPR, 2020.
> 4. Mo, Ningkai, et al. "ES6D: A computation-efficient and symmetry-aware 6D pose regression framework." CVPR, 2022.
> 5. Mitra, Sinjini, and Yanxi Liu. "Local facial asymmetry for expression classification." CVPR, 2004.
> 6. Yu, Chen-Ping, et al. "3D blob-based brain tumor detection and segmentation in MR images." IEEE ISBI, 2014.
> 7. Yang, Mingkun, et al. "Symmetry-constrained rectification network for scene text recognition." CVPR, 2019.
> 8. Patel, Diptiben, Rajendra Nagar, and Shanmuganathan Raman. "Reflection symmetry-aware image retargeting." Pattern Recognition Letters 125 (2019): 179-186.

---

> ### Author Response · Authors · 2024-11-25
> **Gentle reminder for rebuttal review to Reviewer YGvV**
>
> Dear Reviewer YGvV,
>
> We thank you again for your valuable feedback. We have carefully addressed your questions and concerns in our responses. As the rebuttal period ends in two days, we would appreciate your review of our responses. We would be happy to provide any additional clarification if needed. Thank you for your time and effort in reviewing our paper.

---

> ### Author Response · Authors · 2024-12-02
> **Kind reminder: Rebuttal review closing tomorrow to Reviewer YGvV**
>
> Dear Reviewer YGvV
>
> Thank you for your valuable feedback on our paper. We previously submitted our responses to your thoughtful feedback 12 days ago. As the rebuttal period closes tomorrow, we wanted to gently follow up to ensure you have the opportunity to review our responses. Your insights are valuable to us, and we want to make sure we have adequately addressed all your concerns. If you need any clarification or have additional questions, we would be more than happy to provide a prompt response. We greatly appreciate your time and dedication to reviewing our work. Thank you for your consideration.
>
> Best regards,
>
> Authors

---

### Official Review · Reviewer_72W3 · 2024-11-03

**Soundness:** 3
**Presentation:** 2
**Contribution:** 3
**Rating:** 6
**Confidence:** 5

**Summary:**

This paper presents a novel axis-level reflectional symmetry detection network that leverages dihedral group-equivariant representations to improve the detection of symmetry axes in images. The authors introduce an orientational anchor expansion method for fine-grained, rotation-equivariant analysis across multiple orientations, enhancing the model's ability to detect diverse symmetry patterns. They also develop a reflectional matching module using multi-scale kernels to capture reflectional correlations across different receptive fields, improving robustness. Extensive experiments demonstrate that the proposed method outperforms existing pixel-level approaches in challenging scenarios, establishing a new benchmark in reflectional symmetry detection. The work offers a fresh perspective and significant contributions to the field of symmetry detection.

**Strengths:**

* The paper introduces an innovative axis-level reflectional symmetry detection method based on dihedral group-equivariant representations.
* The proposed orientational anchor expansion and reflectional matching modules effectively enhance the model's detection capabilities across various orientations and scales.
* The method demonstrates strong robustness and generalization in complex real-world scenarios.
* The paper provides clear explanations of complex concepts and methodologies, aiding reader comprehension.

**Weaknesses:**

* The implementation details for orientational anchors could be expanded to clarify their integration within the broader architecture and their impact on computational efficiency.
* While multi-scale reflectional matching is beneficial, further analysis on the trade-off between accuracy and computational overhead would improve the study.
* The model’s applicability to continuous symmetries, such as ellipses or curved patterns, is limited, which may constrain its use in certain symmetry-dense applications.
* The dependency on pre-defined kernels in multi-scale matching might limit adaptability to unknown scales or orientations in real-time applications.
* The paper lacks evaluation of the method's generalization performance on different datasets, which could limit its applicability to other scenarios.

**Questions:**

* The implementation details for orientational anchors could be expanded to clarify their integration within the broader architecture and their impact on computational efficiency.
* The model’s applicability to continuous symmetries, such as ellipses or curved patterns, is limited, which may constrain its use in certain symmetry-dense applications.
* The paper lacks evaluation of the method's generalization performance on different datasets, which could limit its applicability to other scenarios.

---

> ### Author Response · Authors · 2024-11-20
> **Response to reviewer 72W3 (1/3)**
>
> >**W1/Q1:**
> >The implementation details for orientational anchors could be expanded to clarify their integration within the broader architecture and their impact on computational efficiency.
>
> We appreciate the reviewer’s request to clarify the implementation details for orientational anchors and their impact on computational efficiency. Below are detailed explanations addressing these points:
>
> **Implementation details:**
> The orientational anchor mechanism (Sec. 4.2) leverages the channel (group) dimension of the score map, dedicating one score map to each orientation bin in the detection head.
> 1. **Without Orientational Anchors**
> The final score map in Equation 15 is pooled into a single channel, meaning the model predicts a single midpoint score, length, and orientation offset within the range [-π,π] for an (H,W) map.
>
> 2. **With Orientational Anchors**:
> The network predicts N anchor sets of (H,W) maps, where each set corresponds to midpoint scores, lengths, and orientation offsets within a narrower range [-π/N,π/N). This design allows for finer-grained orientation predictions while keeping the core detection head operations (based on one-stage object detection frameworks) consistent. Moreover, since we use each channel of the orientation dimension as an orientational anchor, each anchor naturally captures the response from different orientations, which aligns with our motivation of dividing anchors by the pointing orientation of each anchor.
>
> We provide additional clarification about orientational anchors and their integration within the broader architecture in supplementary material (Sec. D, Fig.7). We illustrate how each group dimension channel works as an anchor using arrows that correspond to specific orientations, demonstrating how these anchors capture lines with different orientations.
>
> **Computational efficiency:**
> The use of orientational anchors provides substantial accuracy improvements, as highlighted in Table 1 of the main text.
> | Method | sAP@5 | sAP@10 | sAP@15 | sAP_img@5 | sAP_img@10 | sAP_img@15 |
> |---------|--------|---------|---------|------------|-------------|-------------|
> | Without Orientational Anchor (Sec. 4.1) | 4.8 | 7.9 | 10.0 | 11.2 | 14.3 | 16.7 |
> | With Orientational Anchor (Sec. 4.2) | 17.3 | 20.8 | 22.6 | 24.6 | 30.1 | 32.0 |
>
> The computational overhead remains minimal because the main difference lies in the pooling operation, which has negligible impact on GPU computation.
> 1. Training Time: 9.5 hours.
> 2. Memory: 14 GB per GPU (total 56 GB for 4 GPUs).
> 3. Computational Cost: 39.5 GFlops.
>
> The orientational anchors significantly enhance accuracy with minimal computational impact, making them a valuable enhancement to our architecture.
>
> >**W2:**
> >While multi-scale reflectional matching is beneficial, further analysis on the trade-off between accuracy and computational overhead would improve the study.
>
> W2A. Thank you for raising this important point about the computational trade-offs.
>
> | Feature || Matching kernel | | Time(h) | Memory(GB) Per/Total | GFLOPs | | sAP | |
> |---------|:--------------:|:-:|:-:|:-------:|:------------------:|:-------:|:---:|:-:|:-:|
> | | 1×1 | 3×3 | 5×5 | | | | @5 | @10 | @15 |
> | **F** | - | - | - | 9.5 | 14 / 56 | 39.5 | 17.3 | 20.8 | 22.6 |
> | [**F**, **H**] | ✓ | | | 13.0 | 22 / 88 | 58.5 | 19.0 | 22.4 | 23.8 |
> | [**F**, **H**] | ✓ | ✓ | | 14.5 | 26 / 104 | 81.2 | 18.5 | 22.4 | 23.7 |
> | [**F**, **H**] | ✓ | ✓ | ✓ | 17.0 | 30 / 120 | 102.5 | 19.7 | 23.9 | 25.7 |
>
> To analyze this trade-off systematically, we calculated the computational costs (Columns 5-7) and compared them with the performance metrics from our main experiments (Columns 8-10). As shown in Table C.5, the computational requirements increase progressively with kernel size: from 39.5 GFLOPs (without matching) to 58.5 GFLOPs (1×1), 81.2 GFLOPs (3×3), and 102.5 GFLOPs (5×5). This is accompanied by corresponding increases in GPU memory usage (from 56GB to 120GB total) and training time (from 9.5h to 17h).
>
> According to the table, this computational investment yields meaningful performance gains. The addition of matching features improves sAP from 17.3/20.8/22.6 (base features only) to 19.0/22.4/23.8 with 1×1 kernels, and further to 19.7/23.9/25.7 with the full multi-scale approach. Notably, while the computational cost increases by 2.6× from base to full multi-scale (39.5 to 102.5 GFLOPs), we observe consistent performance improvements across all metrics (+2.4, +3.1, and +3.1 for sAP@5, @10, and @15 respectively). This suggests that despite the increased computational overhead, the multi-scale reflectional matching strategy provides a reasonable trade-off between computational cost and detection performance.

---

> ### Author Response · Authors · 2024-11-20
> **Response to reviewer 72W3 (2/3)**
>
> >**W3/Q2:**
> >The model's applicability to continuous symmetries, such as ellipses or curved patterns, is limited, which may constrain its use in certain symmetry-dense applications.
>
> Thank you for the insightful question regarding the model’s applicability to continuous or curved symmetries.
>
> **Continuous Symmetries (e.g., Ellipses):**
> In DENDI, ellipse-shaped annotations represent circles that appear as ellipses due to viewpoint changes. These ellipses can be considered as having an infinite number of axes sharing the same center. To address this, we could either add a dedicated branch for ellipse-axis segmentation, similar to EquiSym, or merge multiple overlapping axes into a single ellipse mask. However, our current model is limited to 8 rotational anchors and is specifically trained on datasets including octagons. This constraint may cause ambiguity when distinguishing between circles, ellipses, and octagons. Expanding the model to effectively handle continuous symmetries represents a promising direction for future work.
>
> **Curved Symmetries:**
> Early works such as [1,2] explored curved symmetries, which later developed into the specialized task of skeleton detection. These approaches typically generate labels through post-processing of segmentation outputs and are evaluated on benchmarks such as SYMMAX [2], WH-SYMMAX [3], SK506 [5], Sym-PASCAL [7], and SK-LARGE [6]. Skeleton detection techniques include pixel-based regression using multiple-instance learning [2,3] and deep learning-based segmentation methods for detecting skeletons at various scales [4-8]. While these tasks are closely related with reflectional symmetry detection, both curved symmetry detection and skeleton detection lie beyond the scope of our current study.
>
> **References:**
> 1. Lee, Seungkyu, and Yanxi Liu. "Curved glide-reflection symmetry detection." IEEE TPAMI, 34.2 (2011).
> 2. Tsogkas, Stavros, and Iasonas Kokkinos. "Learning-based symmetry detection in natural images." ECCV, 2012.
> 3. Wei Shen, Xiang Bai, Zihao Hu, and Zhijiang Zhang. "Multiple instance subspace learning via partial random projection tree for local reflection symmetry in natural images." Pattern Recognition, 2015.
> 4. Teo, Ching L., Cornelia Fermuller, and Yiannis Aloimonos. "Detection and segmentation of 2D curved reflection symmetric structures." ICCV, 2015.
> 5. Shen, Wei, et al. "Object skeleton extraction in natural images by fusing scale-associated deep side outputs." CVPR, 2016.
> 6. Wei Shen, Kai Zhao, Yuan Jiang, Yan Wang, Xiang Bai, and Alan Yuille. "DeepSkeleton: Learning multi-task scale-associated deep side outputs for object skeleton extraction in natural images." IEEE TIP, 2017.
> 7. Ke, Wei, et al. "SRN: Side-output residual network for object symmetry detection in the wild." CVPR, 2017.
> 8. Wang, Yukang, et al. "DeepFlux for skeletons in the wild." CVPR, 2019.
>
> >**W4:**
> >The dependency on pre-defined kernels in multi-scale matching might limit adaptability to unknown scales or orientations in real-time applications.
>
> We appreciate the reviewer’s insightful comment regarding the dependency on pre-defined kernels in multi-scale matching and its potential limitations. Below, we address the concerns raised:
>
> 1. **Unknown Orientation of the Reflection Axis:**
> To handle unknown orientations, the matching grid is computed in different directions (Sec. 4.3), and each orientation bin produces predictions (Sec. 4.2). Additionally, this process is rotation-equivariant, making it robust to rotation variations and adaptable to diverse orientations in real-time applications.
>
> 2. **Unknown Object Scale:**
> To address unknown scales, we perform multi-scale matching and concatenate the results, allowing the subsequent layer to learn and select the appropriate scale. This approach ensures adaptability to objects of varying sizes.
>
> 3. **Scale Variance Between Matched Features:**
> One aspect not currently addressed is the scale variance between the two matched features, which is a significant issue in visual correspondence problems. However, in reflection symmetry detection, this challenge is less severe since the matched keypoints belong to the same reflection-symmetric object. While viewpoint variations might alter the scale of matching counterparts, the impact is relatively minor compared to general visual correspondence tasks.
>
>
> You can also refer to Figure 3, which shows that our method captures axes with varying orientations and scales. We thank the reviewer for highlighting this issue and acknowledge it as an important area for future work to further enhance adaptability and robustness.

---

> ### Author Response · Authors · 2024-11-20
> **Response to reviewer 72W3 (3/3)**
>
> >**W5/Q3:**
> >The paper lacks evaluation of the method's generalization performance on different datasets, which could limit its applicability to other scenarios.
>
> We appreciate the reviewer’s comment regarding the evaluation of our method's generalization performance. To address this, we have expanded our evaluation to include additional results on multiple datasets, as detailed in the supplemental material(C.1):
>
> **Evaluation on LDRS[1] Dataset:**
> - Our method achieved state-of-the-art performance with an F1-score of 0.434.
> - For comparison, the previous best methods, PMCNet(Seo et. al. 2021) and EquiSym (Seo et. al. 2022), achieved F1-scores of 0.373 and 0.400 (reproduced), respectively.
>
> **Evaluation on SDRW[2] Dataset:**
> - Our method achieved the second-best performance with an F1-score of 0.683, closely following PMCNet(Seo et. al. 2021), which achieved 0.686.
>
> | Method | DENDI | LDRS | SDRW |
> |--------|-------|------|------|
> | PMCNet | 32.6 | 37.3 | **68.8** |
> | EquiSym | 36.7 | 40.0 | 67.5 |
> | Ours | **37.2** | **43.4** | 68.3 |
>
> **References**
> 1. Seo, Ahyun, Woohyeon Shim, and Minsu Cho. "Learning to discover reflection symmetry via polar matching convolution." Proceedings of the IEEE/CVF international conference on computer vision. 2021.
> 2. Liu, Jingchen, et al. "Symmetry detection from realworld images competition 2013: Summary and results." Proceedings of the IEEE conference on computer vision and pattern recognition workshops. 2013.

---

> ### Author Response · Authors · 2024-11-25
> **Gentle reminder for rebuttal review to Reviewer 72W3**
>
> Dear Reviewer 72W3,
>
> We thank you again for your valuable feedback. We have carefully addressed your questions and concerns in our responses. As the rebuttal period ends in two days, we would appreciate your review of our responses. We would be happy to provide any additional clarification if needed. Thank you for your time and effort in reviewing our paper.

---

> > ### Comment · Reviewer_72W3 · 2024-11-29
> >
> > Thanks for the kindly reply. It addressed my concerns.I keep my scores for the acceptance of this paper.
> > One more question is that please show some possible explanation about why the method got the second-best performance with an F1-score of 0.683 for SDRW?

---

> ### Author Response · Authors · 2024-11-29
> **Official Comment by Authors**
>
> Thank you for raising this important question about our F1-score performance on the SDRW dataset.
> Previous pixel-level approaches (based on heat map prediction) predict regions where symmetry axes are likely to exist, rather than detecting the axes themselves, as they assign each pixel a probability of being part of an axis region. Accordingly, the ground truth for these axes is provided as an 'axis-likely region', which is essentially a padded line. In contrast, our method is specifically designed to detect individual axes directly, not regions.
>
> The key factor in this comparison lies in how we determine true positives during evaluation. The maximum distance parameter (shown as distance threshold in the table below), which is multiplied by the image diagonal length, determines whether a predicted pixel is considered a correct match to the ground truth. With a generous threshold of 0.01, even blurry predictions from heat map-based methods can be considered correct matches. However, as we make this criterion more strict by reducing the threshold, our method's precise axis detection shows superior performance over PMCNet's region-based predictions.
>
> | Distance threshold (× image diagonal) | PMCNet (F1-score) | Ours (F1-score) | Diff (Ours - PMCNet) |
> |----------------------------------|--------|------|---------------------|
> | 0.010 | 0.688 | 0.683 | -0.005 |
> | 0.009 | 0.678 | 0.675 | -0.003 |
> | 0.008 | 0.657 | 0.661 | +0.004 |
> | 0.007 | 0.626 | 0.644 | +0.018 |
> | 0.006 | 0.588 | 0.619 | +0.031 |
> | 0.005 | 0.543 | 0.557 | +0.014 |
> | 0.004 | 0.468 | 0.493 | +0.025 |
> | 0.003 | 0.382 | 0.399 | +0.017 |
> | 0.002 | 0.279 | 0.287 | +0.008 |
> | 0.001 | 0.133 | 0.145 | +0.012 |
>
> As shown in the table, when the matching criterion becomes more strict (lower distance threshold), our method consistently outperforms PMCNet. This demonstrates that our approach achieves more precise axis localization, which is crucial for practical applications requiring precise symmetry axis detection.
>
> Thank you for your quick response and positive feedback regarding how we addressed your major concern. We're happy to hear that this resolves your primary issue. If you have any remaining concerns that might make you hesitant to increase the score from 6, we welcome further discussion to address them.

---

### Official Review · Reviewer_qQr5 · 2024-11-07

**Soundness:** 3
**Presentation:** 2
**Contribution:** 1
**Rating:** 5
**Confidence:** 5

**Summary:**

This paper presents a reflection symmetry detection system with matched, multiscale kernels rotationally equivariant network.  The work uses equivalent networks to allow symmetries to be detected at rotations.   The authors also use a fiber based approach to directly find the symmetry rather than first predicting a heatmap for the symmetry.

**Strengths:**

The paper is using rotational equivariant networks to improve symmetry detection.  It’s an interesting approach (though needs to be sufficiently distinguished from others in the field).

This paper is clearly written and I can follow the logic on what they are trying to do.

I appreciate the approach with fibers and it seems interesting to not use a dense approach.    I like the difference in the approach and would like to see more of that with different backbones since I think it would work even better.

**Weaknesses:**

Major:
For group representation and a longer background of symmetry detection and needs to be cited here is Computational symmetry in computer vision and computer graphics by Yanxi Liu et al.  2010

The evaluation only compares against a recent method and doesn’t go back to any of the previous methods (check out Funk et al 2017 for a list of methods where most are freely available online).  They are used in the papers the authors compared with: Seo, Ahyun, Woohyeon Shim, and Minsu Cho. "Learning to discover reflection symmetry via polar matching convolution." Proceedings of the IEEE/CVF international conference on computer vision. 2021.  In addition, why have you deviated from the standard precision recall with F1 marked curve like the previously mentioned papers?  That is really useful to understand how your metric compares to others.

This paper (which cited thoughout the paper), also uses equivariant networks for both rotation and reflection symmetry detection.   Seo 2021 and 2022 both use equivariant kernels with Seo 2022 uses rotation group equivalents.  The main difference between the group equivalence of the other papers, at least that I understand, is that the group d8 is used rather than a special Euclidean symmetry group out of the 17.  I'm only referring to the difference in the equivariant and not other differences in the approach.


Missing citations
Gens, R. and Domingos, P. Deep Symmetry Networks. NeurIPs, 2014.  proposed equivariant convolutional arch that needs to be cited and compared with.
Rotationally-invariant CCNs - Dieleman, Sander, Kyle W. Willett, and Joni Dambre. "Rotation-invariant convolutional neural networks for galaxy morphology prediction." Monthly notices of the royal astronomical society 450, no. 2 (2015): 1441-1459.

Authors should mention that this is equivariant NN for just 2D data or other papers such as “Equivariant Multi-View Networks. Carlos Esteves et al.  ICCV 2019” should be cited.

A figure to help understand Sections 3 and 4 would be helpful for understanding what you are getting at here visually.  There is a lot of text and equations and I think a figure to get at the expansion of fibers and how the author sare using symmetry groups would be a big help.


Minor:
First paragraph needs citations.  You can’t just state facts in a paper without a citation on symmetry being a fundamental concept.  You can go back to Gestalt theory or how symmetry detection is prevalent in the animal kingdom but cite it.

In Figure 1, (a) (b)... needs to be labeled in the image.  This is hard to follow

**Questions:**

The authors are basing this on Cohen et al.’s work for equivariant cnns and I’m not sure how is this different?  Those filters are already rotational equivariant based on the symmetry groups they represent.

“Lenc & Vedaldi (2015) show that the AlexNet CNN (Krizhevsky et al., 2012) trained on imagenet spontaneously learns representations that are equivariant to flips, scaling and rotation.” from Group Equivariant Convolutional Network.  Why would this approach be necessary for rotational symmetry invariance for reflection symmetry detection?

How large is the kernel size?  If using too small a size, how can it be 8-fold symmetric?

How well does this objective function work on non-changed neural networks?  What about modern networks like Convnext?

Line 236: why D8 and not some other amount of rotations?

---

> ### Author Response · Authors · 2024-11-20
> **Response to reviewer qQr5 (1/6)**
>
> > **Major W1:**
> >For group representation and a longer background of symmetry detection and needs to be cited here is Computational symmetry in computer vision and computer graphics by Yanxi Liu et al. 2010
>
> We appreciate the reviewer's suggestion to include Liu et al.'s comprehensive 2010 survey on computational symmetry. We have added this important reference in our introduction section(line 31) to better contextualize our work within the broader field of computational symmetry.
>
> > **Major W2:**
> >The evaluation only compares against a recent method and doesn't go back to any of the previous methods (check out Funk et al 2017 for a list of methods where most are freely available online). They are used in the papers the authors compared with: Seo, Ahyun, Woohyeon Shim, and Minsu Cho. "Learning to discover reflection symmetry via polar matching convolution." Proceedings of the IEEE/CVF international conference on computer vision. 2021. In addition, why have you deviated from the standard precision recall with F1 marked curve like the previously mentioned papers? That is really useful to understand how your metric compares to others.
>
> Based on the feedback, we have expanded our evaluation and included additional results in the supplemental material(C.1, C.2):
>
> **Evaluation on Standard Benchmarks:**
>
> **LDRS Dataset(Supp C.1):**
> - Achieved state-of-the-art performance with an F1-score of 0.434
> - Previous best results: PMCNet[2] (0.373) and EquiSym[3] (0.400, reproduced)
> - Precision-recall (PR) curves for this dataset are included in the supplemental material
>
> **SDRW Dataset(Supp C.1, C.2):**
> - Achieved the second-best performance with an F1-score of 0.683, close to the best result by PMCNet (0.686)
> - PR curves are provided, including comparisons with other prior methods such as PMCNet[2], SymResNet[1], LE[7], MIL[6], FSDS[4], and SRF[5]
>
> | Method | DENDI | LDRS | SDRW |
> |--------|-------|------|------|
> | PMCNet | 32.6 | 37.3 | **68.8** |
> | EquiSym | 36.7 | 40.0 | 67.5 |
> | Ours | **37.2** | **43.4** | 68.3 |
>
> **Precision-Recall Curves with F1 Marked(Supp C.1, C.2):**
> - We agree with the reviewer that precision-recall curves are essential for understanding comparative performance. A new PR curve for our results on the DENDI dataset, corresponding to Table 4, has been added to the supplemental material.
> - Regarding the PR curves for the DENDI and LDRS datasets, our method demonstrates relatively higher precision but lower recall compared to other approaches. This behavior arises from our method's post-processing steps, including Non-Maximum Suppression (NMS) and score-based thresholding, applied to the detected lines. In contrast, pixel-level prediction methods (e.g., heat-map-based approaches) achieve higher recall at the lowest thresholds by designating all pixels on the map as predictions.
> - Moreover, the inconsistency in the low recall region is due to the fundamental difference between our axis prediction task and heatmap-based pixel prediction tasks. Unlike heatmap-based methods, where predictions are distributed across all pixels, our method predicts entire lines by estimating the line midpoint and associated score. Consequently, a change in the score of a single line midpoint affects the state of all pixels corresponding to that line. This characteristic leads to higher variability in precision at higher thresholds, where recall is low.
> - We believe these additional evaluations provide a more comprehensive and transparent comparison of our method with previous works. Furthermore, while existing heatmap-based methods face challenges in converting their predictions to explicit axis representations, our approach directly provides more precise and detailed information about symmetry axes, offering a more sophisticated solution for symmetry detection tasks.
>
> | Method | Method Score |
> |--------|-------------|
> | SRF[5] | 0.19 |
> | FSDS[4] | 0.21 |
> | MIL[6] | 0.23 |
> | Sym-VGG[1] | 0.4 |
> | LE[7]| 0.4 |
> | Sym-ResNet[1] | 0.55 |
> | EquiSym[3] | 0.675 |
> | Ours | 0.683 |
> | PMCNet[2] | 0.688 |
>
> **References:**
> 1. Funk, Christopher, and Yanxi Liu. "Beyond planar symmetry: Modeling human perception of reflection and rotation symmetries in the wild." ICCV 2017
> 2. Seo, Ahyun, et al. "Learning to discover reflection symmetry via polar matching convolution." ICCV 2021
> 3. Seo, Ahyun, et al. "Reflection and rotation symmetry detection via equivariant learning." Proceedings of the IEEE/CVF conference on computer vision and pattern recognition, 2022
> 4. Shen, Wei, et al. "Object skeleton extraction in natural images by fusing scale-associated deep side outputs." CVPR 2016
> 5. Teo, Ching L., Cornelia Fermuller, and Yiannis Aloimonos. "Detection and segmentation of 2D curved reflection symmetric structures." ICCV 2015
> 6. Tsogkas, Stavros, and Iasonas Kokkinos. "Learning-based symmetry detection in natural images." ECCV 2012
> 7. Gareth Loy et.al. "Detecting symmetry and symmetric constellations of features." ECCV  2006.

---

> ### Author Response · Authors · 2024-11-20
> **Response to reviewer qQr5 (2/6)**
>
> >**Major W3:**
> >This paper (which cited throughout the paper), also uses equivariant networks for both rotation and reflection symmetry detection. Seo 2021 and 2022 both use equivariant kernels with Seo 2022 uses rotation group equivalents. The main difference between the group equivalence of the other papers, at least that I understand, is that the group d8 is used rather than a special Euclidean symmetry group out of the 17. I'm only referring to the difference in the equivariant and not other differences in the approach.
>
>
>  We appreciate the reviewer's thoughtful analysis and would like to clarify that Seo2021 did not employ equivariant kernels or equivariant networks.
>
> While our work builds upon previous contributions in equivariant networks, we would like to clarify several key distinctions in our approach:
> First, our work differs from previous approaches (Funk2017, Seo2021, Seo2022) by maintaining rotational equivariance while predicting axes through a CAL (Center-Angle-Length) based detection rather than traditional heatmap-based prediction. This naturally leads us to consider different equivariance properties for each component: length needs to be rotation-invariant, while orientation value and midpoint position have to maintain rotation-equivariance. We address these challenges through our novel orientation anchor approach (Section 4.2).
>
> **Regarding the comparison with Seo's work:**
> - While Seo2021 attempts to handle rotational variations through polar matching, their method differs in that they do not employ equivariant networks or kernels in feature extraction. Their non-equivariant nature is directly acknowledged in their paper: "Since CNNs are neither invariant nor equivariant to rotation and reflection, PMC may still have difficulty in learning to detect symmetry." (Section 3.2). Their reliance on spatial matching with non-equivariant descriptors means the network lacks rotational equivariance at both local and global levels. As demonstrated in our supplementary material (C.3), the performance gap between non-equivariant and equivariant networks clearly validates the effectiveness of combining our contributions with equivariant architectures. Specifically, our equivariant network using equivariant ResNet as backbone achieves superior performance (19.7/23.9/25.7) in sAP @5, @10, and @15 compared to non-equivariant ResNet implementations using either single anchor (4.7/7.9/10.1) or orientational anchor (15.6/19.2/20.1) approaches.
>
> | Backbone | Method | sAP@5 | sAP@10 | sAP@15 |
> |----------|---------|-------|--------|--------|
> | ResNet-34 | Single Anchor | 4.7 | 7.9 | 10.1 |
> | ResNet-34 | Orientational Anchor | 15.6 | 19.2 | 20.1 |
> | Ours (Group-CNN) | Orientational Anchor | **19.7** | **23.9** | **25.7** |
>
> - While Seo2022 maintains group dimension in feature maps (R^{C|C_N|×H×W}) before the final prediction head, similar to our approach, there is a crucial difference in how we handle the orientation dimension. Instead of using pooling techniques that collapse the group dimension and only involve spatial dimension for equivariance constraint, we explicitly preserve and utilize the group dimension through independent orientational anchors. This allows us to:
>
>    1. Independently leverage the information contained in each group dimension channel, preserving orientation-specific features
>    2. Utilize each group dimension as a dedicated anchor for predicting axes oriented near its characteristic direction
>    3. Maintaining rotation equivariance of orientation output by computing orientation offsets relative to each anchor's center angle
>
> The effectiveness of this approach is demonstrated in Table 1 (rows 1 and 2). While both approaches maintain equivariance, our orientation anchor-based method (row 2: 17.3/20.8/22.6) significantly outperforms the conventional approach (row 1: 4.8/7.9/10.1) in sAP @5, @10, and @15, showing that preserving group dimension information through anchors not only maintains equivariance but also leads to more accurate axis-level predictions.

---

> ### Author Response · Authors · 2024-11-20
> **Response to reviewer qQr5 (3/6)**
>
> >**Major W4:**
> >Missing citations Gens, R. and Domingos, P. Deep Symmetry Networks. NeurIPs, 2014. proposed equivariant convolutional arch that needs to be cited and compared with. Rotationally-invariant CCNs - Dieleman, Sander, Kyle W. Willett, and Joni Dambre. "Rotation-invariant convolutional neural networks for galaxy morphology prediction." Monthly notices of the royal astronomical society 450, no. 2 (2015): 1441-1459
>
> Thank you for bringing attention to these important references. We have indeed included both works in our paper: Gens and Domingos' seminal work (2014) is acknowledged in our introduction(line 41) as one of the first equivariant architectures for handling symmetries in CNNs. Both this work and Dieleman et al.'s domain-specific approach are discussed in our Related Work section under paragraph Equivariant neural networks(line 105). Furthermore, we specifically compare our approach with Dieleman et al.'s work in our Method section (line 194), noting how Dieleman et al. deal with rotational invariance for a specific task by transforming input images explicitly and ensembling their predictions.
>
> >**Major W5:**
> >Authors should mention that this is equivariant NN for just 2D data or other papers such as “Equivariant Multi-View Networks. Carlos Esteves et al. ICCV 2019” should be cited.
>
> We appreciate the reviewer's suggestion. We have clarified that our approach focuses on 2D data by explicitly mentioning that our method operates on reflectional symmetry patterns from an “image” (line 32) and further emphasized this by specifying “ 2D reflectional symmetry detection" and "2D feature extraction" (line 52, line 58). These modifications make it clearer that our network's scope is specifically designed for and operates on 2D image data.
>
> >**Major W6:**
> >A figure to help understand Sections 3 and 4 would be helpful for understanding what you are getting at here visually. There is a lot of text and equations and I think a figure to get at the expansion of fibers and how the author share using symmetry groups would be a big help.
>
> We appreciate the reviewer's suggestion regarding the need for visual clarification in Sections 3 and 4. We have added a new figure in Supplementary Material D (Fig. 7) featuring a D8 group example. This figure visualizes both the construction of orientational anchors with respect to group dimensions and the step-by-step process of multi-scale expansion through fiber expansion. This additional visualization provides further clarity to our theoretical framework and addresses some gaps in our previous explanations.
>
> >**Minor W1:**
> >First paragraph needs citations. You can't just state facts in a paper without a citation on symmetry being a fundamental concept. You can go back to Gestalt theory or how symmetry detection is prevalent in the animal kingdom but cite it.
>
> Thank you for bringing our attention to the need for citations regarding fundamental concepts about symmetry. We have thoroughly addressed this by adding relevant citations from foundational works across different domains: We have cited [1] and [2] (line 28) to establish symmetry as a fundamental concept, drawing from Gestalt psychology principles and empirical studies of symmetry perception. To demonstrate the prevalence of symmetry across different domains, we referenced [3] and [4] (line 29), which provide evidence from evolutionary biology and animal cognition studies. Additionally, we included [5] (line 30) to support our statement about human symmetry recognition capabilities, as this work provides comprehensive analysis of visual symmetry detection mechanisms. These citations now support our introductory statements about symmetry's fundamental nature and its widespread occurrence in both natural and artificial environments, as well as human perception capabilities.
>
> **References:**
> 1. Wertheimer, M. "Laws of organization in perceptual forms." In A source book of Gestalt psychology, pages 71-88. Routledge & Kegan Paul, London, 1938.
> 2. Tyler, C. W. "Empirical aspects of symmetry perception." Spatial Vision, 9(1):1-7, 1995.
> 3. Møller, A. P. and Thornhill, R. "Bilateral symmetry and sexual selection: a meta‐analysis." The American Naturalist, 151(2):174-192, 1998.
> 4. Giurfa, M., Eichmann, B., and Menzel, R. "Symmetry perception in an insect." Nature, 382(6590):458-461, 1996.
> 5. Wagemans, J. "Detection of visual symmetries." Spatial vision, 9(1):9-32, 1995.

---

> ### Author Response · Authors · 2024-11-20
> **Response to reviewer qQr5 (4/6)**
>
> >**Minor W2:**
> >In Figure 1, (a) (b)... needs to be labeled in the image. This is hard to follow.
>
> Thank you for pointing out this issue. We have improved Figure 1's clarity by adding alphabetical labels and distinct colors for better readability. We have also refined the visualization of ambiguous components to ensure better understanding.
>
> >**Q1:**
> >The authors are basing this on Cohen et al.'s work for equivariant cnns and I'm not sure how is this different? Those filters are already rotational equivariant based on the symmetry groups they represent.
>
>
> Thank you for raising this important point about group convolutions. While our work builds upon Cohen et al.'s framework, we advance it with key contributions including orientational anchors and group-aware reflectional matching. Our orientational anchors leverage the inherent orientation-specific responses in group CNNs' channels, while our group-aware matching maintains equivariance throughout the pipeline.
>
> Experimental results in Table 1 validate this approach. Incorporating all our contributions (row 4: 19.7/23.9/25.7) significantly outperforms the baseline using only Cohen et al. 's group convolution (row 1: 4.8/7.9/10.1) in sAP @5, @10, and @15. Interestingly, as shown in Table 6 of Supplementary material, this baseline performance is comparable to non-equivariant networks(4.7/7.9/10.1), suggesting that merely having rotational equivariance through group convolutions is insufficient. This highlights that our key contribution lies not in the use of equivariant networks per se, but in our novel symmetry detection framework that extensively leverages equivariant properties.
>
> | Backbone | Method | sAP@5 | sAP@10 | sAP@15 |
> |----------|---------|-------|--------|--------|
> | ResNet-34 | Single Anchor | 4.7 | 7.9 | 10.1 |
> | Group-CNN | Single Anchor| 4.9 | 7.9 | 10.0 |
> | Ours (Group-CNN) | Orientational Anchor | **19.7** | **23.9** | **25.7** |
>
> >**Q2:**
> >"Lenc & Vedaldi (2015) show that the AlexNet CNN (Krizhevsky et al., 2012) trained on imagenet spontaneously learns representations that are equivariant to flips, scaling and rotation." from Group Equivariant Convolutional Network. Why would this approach be necessary for rotational symmetry invariance for reflection symmetry detection?
>
> Thank you for your insightful question regarding the necessity of explicitly addressing rotational equivariance for robust reflection symmetry detection. While Lenc & Vedaldi (2015) demonstrate that CNNs, such as AlexNet trained on ImageNet, can naturally learn a certain degree of rotational equivariance, reflection symmetry detection is a task that requires strong dihedral equivariance. In this context, relying on learned, non-explicit equivariance is problematic, as it often fails to fully capture the complex transformation properties needed. Thus, there are key reasons why explicitly enforcing this property using Group Equivariant Convolution is critical:
>
> - Inconsistent Equivariance: Lenc & Vedaldi’s experiments reveal that the naturally learned equivariance is inconsistent across transformations and layers. In their results (refer to Table 2 of their work), performance under 90° rotations shows significant variance depending on the layer, with top-1 error rates ranging from 0.44 to 0.53. Furthermore, Table 3 of their work shows that a subset of feature channels (ranging from 10% to 43%) exhibit invariance properties under 90° rotations.
>
> - Degraded Performance with Larger Rotations: As depicted in Figure 7 of their work, the classification error increases significantly with larger rotation angles. This highlights that the naturally learned rotational equivariance in standard CNNs is unreliable, especially for handling larger angular transformations.
>
> Given these limitations, the mathematically guaranteed equivariance provided by Group convolutions is essential for robust reflection symmetry detection. Explicitly handling rotational symmetry ensures that the model can generalize well across symmetry axes in various orientations, overcoming the inconsistencies and performance drops observed in standard CNNs.
>
> **References:**
> 1. Lenc, Karel, and Andrea Vedaldi. "Understanding image representations by measuring their equivariance and equivalence." Proceedings of the IEEE conference on computer vision and pattern recognition. 2015.

---

> ### Author Response · Authors · 2024-11-20
> **Response to reviewer qQr5 (5/6)**
>
> >**Q3:**
> >How large is the kernel size? If using too small a size, how can it be 8-fold symmetric?
>
>
> In our implementation, we use 3×3 kernels for group convolutions. While it is true that smaller kernels may pose limitations in maintaining perfect 8-fold symmetry, prior works [1, 2] have demonstrated that 3×3 kernels can still perform adequately with D8 group convolutions by applying proper kernel rotation interpolation.
> Building on this, we also observed in our experiments that increasing the kernel size to 5×5 in the matching module led to improved performance (Table 2, 18.5/22.4/23.7 -> 19.7/23.9/25.7 for sAP@5, @10, and @15), likely due to better handling of rotational symmetry. This result aligns with your concern and highlights how we addressed it in our design.
>
> | Method | sAP@5 | sAP@10 | sAP@15 | sAP_img@5 | sAP_img@10 | sAP_img@15 |
> |---------|--------|---------|---------|------------|-------------|-------------|
> | 1x1 kernel | 19.0 | 22.4 | 23.8 | **26.4** | 30.5 | 32.2 |
> | 1x1 + 3x3 + 5x5 kernels| **19.7** | **23.9** | **25.7** | 26.2 | **31.0** | **33.6** |
>
> **References:**
> 1. Cohen, Taco, and Max Welling. "Group equivariant convolutional networks." International conference on machine learning. PMLR, 2016.
> 2. Han, J., Ding, J., Xue, N., & Xia, G. S. (2021). Redet: A rotation-equivariant detector for aerial object detection. In Proceedings of the IEEE/CVF conference on computer vision and pattern recognition (pp. 2786-2795).
>
>
> >**Q4:**
> >How well does this objective function work on non-changed neural networks? What about modern networks like Convnext?
>
> Thank you for raising these important questions about the applicability of our method to non-equivariant and modern architectures. We understand that the suggestion to use ConvNeXt aims to evaluate our method with more powerful encoders. However, it's important to note that from our main experiments, we identified significant overfitting issues with the DENDI dataset when training with unfrozen encoders due to its limited size. Therefore, to maintain consistent experimental settings with our main results and ensure fair comparison, we conducted experiments with ConvNeXt using the same configuration - ImageNet pre-trained encoders in a frozen state. We apologize for not explicitly mentioning the frozen encoder setting in the implementation details section and thank you for bringing this to our attention. We have added this information in (line 404) of our paper.
>
> For these non-equivariant architectures (ResNet-34 and ConvNeXt), we adapted our approach by implementing spatial matching instead of group-aware matching, as they lack a group structure (see Section 5.3: Matching Strategies). To implement orientational anchors, we designed the final head with 4 (#anchor) channels, without considering group dimensions, and applied distinct ground truths to each anchor, where each ground truth corresponds to lines with different orientations.
>
> Under these settings, we report the evaluation results wit sAP @5, @,10 and @15  in Supplementary material (Sec.C.3, Table 6) with a single anchor, ConvNeXt achieved sAP @5, @10, and @15  of 11.1/15.5/17.3 and ResNet-34 achieved 4.7/7.9/10.1. When using orientational anchors (without group dimension), the performance improved significantly: ConvNeXt reached 18.1/21.1/22.2 and ResNet-34 reached 15.6/19.2/20.1. However, these results still fall short of our group-aware approach (19.7/23.9/25.7), demonstrating that the combination of group-aware matching and orientational anchors aligned with group dimension provides superior performance. These results suggest that  the equivariance properties and group-aware design of our approach provide benefits that exceed those of using more modern architectures.
>
> | Backbone | Method | sAP@5 | sAP@10 | sAP@15 |
> |----------|---------|-------|--------|--------|
> | ResNet-34 | Single Anchor | 4.7 | 7.9 | 10.1 |
> | ResNet-34 | Orientational Anchor | 15.6 | 19.2 | 20.1 |
> | ConvNeXt | Single Anchor | 11.1 | 15.5 | 17.3 |
> | ConvNeXt | Orientational Anchor | 18.1 | 21.1 | 22.2 |
> | Ours (Group-CNN) | Orientational Anchor | **19.7** | **23.9** | **25.7** |
>
> While we acknowledge the potential of exploring various backbones, both equivariant and non-equivariant, the current limitation lies in the limited DENDI dataset of 1.6k training examples. Even with extensive data augmentation strategies, we observed significant overfitting issues with any unfrozen backbone architectures. We believe that as larger datasets become available, further investigation of different backbone architectures could provide additional validation of our method's effectiveness.

---

> ### Author Response · Authors · 2024-11-20
> **Response to reviewer qQr5 (6/6)**
>
> >**Q5:**
> >Line 236: why D8 and not some other amount of rotations?
>
> The selection of the D8 group was based on its balanced level of rotational granularity. Among D4, D8, and D16 options, D8's ±22.5° orientation anchors provide sufficient angular resolution, avoiding both the coarse divisions of D4 (±45°) and overly fine granularity of D16 (±11.25°).
>
> Furthermore, considering the trade-off between equivariance and expressibility, we believe D8 is the most appropriate choice for our work. While larger rotation group sizes could theoretically offer finer rotational equivariance, they may increase computational demands or potentially limit network expressiveness with same physical channel numbers. Future advancements in efficient networks handling larger rotation group sizes could potentially enable improved performance with finer rotational granularity.

---

> ### Author Response · Authors · 2024-11-25
> **Gentle reminder for rebuttal review to Reviewer qQr5**
>
> Dear Reviewer qQr5,
>
> We thank you again for your valuable feedback. We have carefully addressed your questions and concerns in our responses. As the rebuttal period ends in two days, we would appreciate your review of our responses. We would be happy to provide any additional clarification if needed. Thank you for your time and effort in reviewing our paper.

---

> ### Author Response · Authors · 2024-12-02
> **Kind reminder: Rebuttal review closing tomorrow to Reviewer qQr5**
>
> Dear Reviewer qQr5,
>
> Thank you for your valuable feedback on our paper. We previously submitted our responses to your thoughtful feedback 12 days ago. As the rebuttal period closes tomorrow, we wanted to gently follow up to ensure you have the opportunity to review our responses. Your insights are valuable to us, and we want to make sure we have adequately addressed all your concerns.
> If you need any clarification or have additional questions, we would be more than happy to provide a prompt response. We greatly appreciate your time and dedication to reviewing our work.
> Thank you for your consideration.
>
> Best regards,
>
> Authors

---

### Author Response · Authors · 2024-11-21
**General Response to All Reviewers**

We sincerely thank the reviewers for taking the time to provide such thoughtful and constructive feedback on our manuscript. We have carefully addressed the comments in our responses, and incorporated additional details into both the main text and supplementary materials, marked in red for easy reference. We would greatly appreciate if the reviewers could verify these modifications.

The key modifications are as follows:
- Added Supplementary Material C.1 about evaluation on SDRW and LDRS datasets (including PR curves)
- Added Supplementary Material C.2 about comparison with previous methods (including PR curves)
- Added Supplementary Material C.3 about ablation studies with different backbones
- Added Supplementary Material C.4 about ablation studies on line segment detection
- Added Supplementary Material C.5 about ablation studies on computational overhead
- Added Supplementary Material D about architecture details and additional figure (Figure 7)
- Added citations (L28, L29, L30, L31, L48, L49, L50, L105, L106, L107, L194)
- Updated Figure 1 with alphabetical notations and revised visual elements, supported by color-coded indications for better clarity
- Added implementation details of frozen backbone (L404)
- Updated terminology from "multi-scale expansion" to "multi-scale matching" to better reflect our method (L342, L426, Figure 1, Table 1)
- Clarify our method is for 2D reflection symmetry detection (L52, 58, 196)

We hope these revisions have helped to clarify our work in response to the reviewers' valuable feedback.

---

### Author Response · Authors · 2024-12-04
**Final Comment by Authors**

Dear Reviewers, Area Chairs, Senior Area Chairs, and Program Chairs.
We sincerely thank all reviewers for their thorough evaluation and constructive feedback throughout the review process, and all the support of Area Chairs, Senior Area Chairs, and Program Chairs for managing the review process. We are pleased that we were able to address most of the reviewers' concerns through our detailed responses:

- In response to Reviewer qQr5, we provided extensive comparison results including PR curves and comprehensive comparisons with previous methods([1, 2, 3, 4, 5, 6]) on multiple datasets (DENDI[1], LDRS[2], and SDRW[7]). We incorporated fundamental symmetry detection works including Gestalt theory and computational symmetry studies([8, 9, 10, 11, 12]), and demonstrated that our method outperforms modern backbones such as ConvNeXt[13] through detailed ablation studies. We clarified our method's contributions from an equivariance perspective through detailed comparisons and analysis, and enhanced the understanding of our method by improving existing figures and adding a new figure illustrating the fiber expansion and orientational anchor mechanisms.

- In response to Reviewer 72W3, we provided a comprehensive analysis of computational efficiency and architecture details, including a new figure demonstrating the integration of orientational anchors within the broader architecture. We also conducted evaluations across multiple datasets (DENDI[1], LDRS[2], and SDRW[7]), with a detailed study on performance characteristics such as precision-recall trade-offs and F1 scores at various distance thresholds.

- In response to Reviewer YGvV, we expanded our evaluation to additional benchmarks (LDRS[2] and SDRW[7]), demonstrated our method's effectiveness on the Wireframe dataset[14], clarified terms about multi-scale matching, and provided examples of practical applications across medical imaging[15], robotics([16, 17]), and computer vision domains([18, 19]).

We believe our paper presents valuable contributions to the field of reflectional symmetry detection, as evidenced by our experimental results and the positive feedback received during the review process.

Thank you for your time and consideration.

Best regards,

Authors


**References**
1. Seo, Ahyun, et al. "Reflection and rotation symmetry detection via equivariant learning." Proceedings of the IEEE/CVF conference on computer vision and pattern recognition, 2022
2. Seo, Ahyun, et al. "Learning to discover reflection symmetry via polar matching convolution." ICCV 2021
3. Shen, Wei, et al. "Object skeleton extraction in natural images by fusing scale-associated deep side outputs." CVPR 2016
4. Teo, Ching L., Cornelia Fermuller, and Yiannis Aloimonos. "Detection and segmentation of 2D curved reflection symmetric structures." ICCV 2015
5. Tsogkas, Stavros, and Iasonas Kokkinos. "Learning-based symmetry detection in natural images." ECCV 2012
6. Gareth Loy et.al. "Detecting symmetry and symmetric constellations of features." ECCV 2006.
7. Liu, Jingchen, et al. "Symmetry detection from realworld images competition 2013: Summary and results." Proceedings of the IEEE conference on computer vision and pattern recognition workshops. 2013.
8. Wertheimer, M. "Laws of organization in perceptual forms." In A source book of Gestalt psychology, pages 71-88. Routledge & Kegan Paul, London, 1938.
9. Tyler, C. W. "Empirical aspects of symmetry perception." Spatial Vision, 9(1):1-7, 1995.
10. Møller, A. P. and Thornhill, R. "Bilateral symmetry and sexual selection: a meta‐analysis." The American Naturalist, 151(2):174-192, 1998.
11. Giurfa, M., Eichmann, B., and Menzel, R. "Symmetry perception in an insect." Nature, 382(6590):458-461, 1996.
12. Wagemans, J. "Detection of visual symmetries." Spatial vision, 9(1):9-32, 1995.
13. Liu, Zhuang, et al. "A convnet for the 2020s." Proceedings of the IEEE/CVF conference on computer vision and pattern recognition. 2022.
14. Huang, Kun, et al. "Learning to parse wireframes in images of man-made environments." Proceedings of the IEEE Conference on Computer Vision and Pattern Recognition. 2018.
15. Yu, Chen-Ping, et al. "3D blob-based brain tumor detection and segmentation in MR images." IEEE ISBI, 2014.
16. Li, Wai Ho, Alan M. Zhang, and Lindsay Kleeman. "Bilateral symmetry detection for real-time robotics applications." The International Journal of Robotics Research 27.7 (2008): 785-814.
17. Mo, Ningkai, et al. "ES6D: A computation-efficient and symmetry-aware 6D pose regression framework." CVPR, 2022.
18. Yang, Mingkun, et al. "Symmetry-constrained rectification network for scene text recognition." CVPR, 2019.
19. Mitra, Sinjini, and Yanxi Liu. "Local facial asymmetry for expression classification." CVPR, 2004.

---

### Meta-Review · Area_Chair_4DkW · 2024-12-17

**Metareview:**

This submission received two negative sores and a positive score after rebuttal. The remaining major concerns are about the weaker performance and unclear explanation of the proposed approach. After carefully reading the paper, the review comments, the AC can not recommend the acceptance of this submission, which is based the fact that the average score is under the threshold bar. The AC also recognizes the contributions confirmed by the reviewers, and encourages the authors to update the paper according to the discussion and submit it to the upcoming conference.

**Additional Comments On Reviewer Discussion:**

After discussion, while reviewer #qQr5 thought that `the authors responded to most of the issues, but keeps a negative rating. Reviewer#YGvV thought that authors had sufficiently addressed most of his concerns. The remaining issue is that when extending the proposed method to ELSD, the performance seems to be weaker than the original method (62.2/66.5/68.3 vs. 64.3/68.9/70.9). To prove the effectiveness of the method in this paper, he thinks that it is necessary to explain the performance when only discarding the offset and centerness branches of ELSD.

---

### Decision · Program_Chairs · 2025-01-22

Reject